# Causally Abstracted Multi-armed Bandits

**Fabio Massimo Zennaro**[1]    **Nicholas Bishop**[2]    **Joel Dyer**[2]    **Yorgos Felekis**[3]    **Anisoara Calinescu**[2]    **Michael Wooldridge**[2]                    **Theodoros Damoulas**[3]

[1]University of Bergen
[2]University of Oxford
[3]University of Warwick

## Abstract

Multi-armed bandits (MAB) and causal MABs (CMAB) are established frameworks for decision-making problems. The majority of prior work typically studies and solves individual MAB and CMAB in isolation for a given problem and associated data. However, decision-makers are often faced with multiple related problems and multi-scale observations where joint formulations are needed in order to efficiently exploit the problem structures and data dependencies. Transfer learning for CMABs addresses the situation where models are defined on *identical* variables, although causal connections may differ. In this work, we extend transfer learning to setups involving CMABs defined on potentially different variables, with varying degrees of granularity, and related via an abstraction map. Formally, we introduce the problem of causally abstracted MABs (CAMABs) by relying on the theory of causal abstraction in order to express a rigorous abstraction map. We propose algorithms to learn in a CAMAB, and study their regret. We illustrate the limitations and the strengths of our algorithms on a real-world scenario related to online advertising.

## 1 INTRODUCTION

*Multi-armed bandit problems* (MABs) provide an established formalism to model real-world decision-making problems where an agent is required to repeatedly take an action whilst managing a trade-off between exploiting current knowledge or exploring new alternatives [Lattimore and Szepesvári, 2020]. The notion of causality has been explicitly introduced in the MAB formulation by modelling the system upon which an agent may act as a *structural causal model* (SCM) [Pearl, 2009]. This has led to the definition of

*causal MABs* (CMABs), where the actions of an agent are identified with causal interventions, and the relation between an action and the outcome of the system is mediated by the SCM [Bareinboim et al., 2015, Lattimore et al., 2016].

Although MABs are often solved in isolation, growing interest has focused on the problem of transferring learned information across MABs, in order to more efficiently solve decision-making problems in different environments [Lazaric, 2012]. Current methods for CMABs exploit common variables and structures between a source model and a target model, in order to transfer information [Zhang and Bareinboim, 2017]. In this work, we study the possibility of transferring information between CMABs that may be defined over related, but different, variables. Specifically, we consider the case where a decision-making problem is modelled through two CMABs at different levels of resolution providing two representations of the same system. This setup mirrors common real-world scenarios where observations and experiments for critical decision-making have been carried out at different scales. For instance, consider the common scenario of optimal advertisement placement: an online company may deploy different protocols to collect click-through data about its customers. These protocols may differ in the variables that are observed due to different technical constraints, legal requirements or management decisions at the time of collection. While data gathered from each protocol may be used to solve isolated CMABs, if the underlying models could be formally related to each other, information may be transferred from one CMAB to the other, thus improving the learning process.

In order to transfer information across CMABs defined over different variables, we rely on the theory of causal abstraction (CA) to relate the causal structure of the CMABs [Rischel, 2020, Zennaro et al., 2023b]. We thus define the causally abstracted MABs (CAMAB) problem, that is, the problem of learning across CMABs by exploiting a known abstraction map. Whereas current approaches to causal transfer require models to be defined on identical variables, our approach overcomes this limitation, and allows us to transfer

samples and aggregate statistics from a low-level model to a high-level model in different settings. We propose algorithms for learning in CAMABs and analyze our results in terms of simple and cumulative regret.

**Contributions.** This paper introduces a framework based on CA for transferring information between CMABs which moves beyond previous works connecting transfer learning and causal inference under the assumption of a fixed graphical structure [Rojas-Carulla et al., 2018, Pearl and Bareinboim, 2011]. We define the CAMAB problem (Sec. 3), present a customized measure for the quality of abstraction (Sec. 4), and discuss a taxonomy of CAMAB settings (Sec. 5). We then study the most representative CAMAB settings: we first derive negative results for an intuitive algorithm transporting the optimal action of the base CMAB (Sec. 5.1) and for an off-policy algorithm transporting the actions taken in the base CMAB (Sec. 5.2); then, we propose and analyze an algorithm based on the transfer of the expected values of the rewards and their upper bounds from the base CMAB (Sec. 5.3). We thus provide a broad overview of different approaches within our framework, and, at the end, we showcase our algorithms on a realistic problem (Sec. 6).

## 2 BACKGROUND

In this section we review the definition of SCMs, integrate them into MABs to define CMABs, and finally formalize the notion of CA between SCMs. We refer the reader to App. B for a discussion of the assumptions in our models.

**Causal Models.** A SCM encodes a causal system defined over a set of variables as follows:

**Definition 2.1** (SCM [Pearl, 2009]). A structural causal model (SCM) is a tuple $\mathcal{M} = \langle \mathcal{X}, \mathcal{U}, \mathcal{F}, \mathbb{P}(\mathcal{U}) \rangle$ where:

- $\mathcal{X} = \{X_1, ..., X_n\}$ is a set of $n$ endogenous variables, each defined on domain $\mathbb{D}[X_i]$;
- $\mathcal{U} = \{U_1, ..., U_n\}$ is a set of $n$ exogenous variables, each defined on domain $\mathbb{D}[U_i]$;
- $\mathcal{F} = \{f_1, ..., f_n\}$ is a set of $n$ measurable functions, one for each endogenous variable $X_i$; each function is defined $f_i : \mathbb{D}[\mathcal{X}] \setminus \mathbb{D}[X_i] \times \mathbb{D}[U_i] \to \mathbb{D}[X_i]$;
- $\mathbb{P}(\mathcal{U})$ is a joint probability distribution over the exogenous variables $\mathcal{U}$.

An SCM $\mathcal{M}$ admits an underlying directed acyclic graph (DAG) $\mathcal{G}_\mathcal{M} = \langle V, E \rangle$, where $V$ is the set of vertices $V = \mathcal{X} \cup \mathcal{U}$ and $E$ is the set of edges $E = \{(V_i, V_j)|V_j \in \mathcal{X}, V_i \in \mathcal{X} \cup \mathcal{U}$ and $V_i$ in domain of $f_j\}$.

A decision-maker may act on an SCM through interventions:

**Definition 2.2** (Intervention [Pearl, 2009]). Given a SCM $\mathcal{M}$, an intervention $\mathrm{do}(\mathbf{X} = \mathbf{x})$, shortened to $\mathrm{do}(\mathbf{x})$ when

clear from the context, is an operator which, for every $X_i \in \mathbf{X} \subseteq \mathcal{X}$ replaces the structural function $f_i$ in $\mathcal{M}$ with the constant $x_i \in \mathbb{D}[X_i]$.

Note that an intervention $\mathrm{do}(\mathbf{x})$ on $\mathcal{M}$ mutilates the graph $\mathcal{G}_\mathcal{M}$ by removing all edges incoming to each variable $X \in \mathbf{X}$ and induces a new probability distribution $\mathbb{P}(\mathcal{X} \mid \mathrm{do}(\mathbf{X} = \mathbf{x}))$ over endogenous variables.

**Causal MABs.** We now combine SCM with MABs following Lattimore et al. [2016]. A CMAB $\mathcal{B}$ is defined by a SCM $\mathcal{M}$ with a predesignated reward variable $Y \in \mathcal{X}$, and a set $\mathcal{A}$ of actions $a_i$ which we equate to a set $\mathcal{I}$ of interventions $\mathrm{do}(\mathbf{x}_i)$ on $\mathcal{M}$. Similarly, we take rewards $g$ generated by the reward process $G_{a_i}$ associated with action $a_i$ to correspond to samples $y$ from $\mathbb{P}(Y \mid \mathrm{do}(\mathbf{x}_i))$. We will assume that $\mathcal{A}$ is finite and that the reward values lie in the interval $[0, 1]$ almost surely. We denote the expected reward for action $a$ by $\mu_a = \mathbb{E}_{Y|a}[Y]$; we also denote the optimal action as $a^* = \mathrm{argmax}_{a \in \mathcal{A}} \mu_a$ and the optimal expected reward as $\mu^* = \max_{a \in \mathcal{A}} \mu_a$. See Tab.1 for an equivalence between SCM and CMAB quantities. A learning agent interacts with $\mathcal{M}$ over $T$ rounds; in each round $t$, it selects and performs action $a^{(t)}$, and collects reward $y^{(t)}$. The goal of the agent is to adopt an intervention policy $\pi^{(t)}(a) = \mathbb{P}(a)$ that minimises a regret-based performance benchmark. We consider two different notions of regret, the first of which is *simple regret*:

$$\bar{R}(T) = \mu^* - \mathbb{E}_{\pi^{(T)}}\left[\mu_{a^{(T)}}\right], \tag{1}$$

where $a^{(T)}$ is an arm chosen by the learner after the time horizon has concluded. MABs evaluated under simple regret are often referred to as pure exploration problems, as the goal of the agent is simply to identify the best arm over the course of the time horizon, rather than play it repeatedly. The second benchmark we consider is *cumulative regret*:

$$R(T) = T\mu^* - \sum_{t=1}^{T} \mathbb{E}_{\pi^{(t)}}\left[\mu_{a^{(t)}}\right], \tag{2}$$

wherein the goal is to maximise its expected reward accumulated over the time horizon w.r.t. the learned policy. A standard algorithm to solve MAB problems is offered by the `UCB` algorithm [Lattimore and Szepesvári, 2020].

**Causal Abstraction.** Finally, we review the notion of abstraction allowing us to relate SCMs.

**Definition 2.3** (Abstraction [Rischel, 2020]). Given two SCMs $\mathcal{M}$ and $\mathcal{M}'$, an abstraction is a tuple $\boldsymbol{\alpha} = \langle V, m, \alpha \rangle$, where:

- $V \subseteq \mathcal{X}$ is a subset of relevant variables in $\mathcal{M}$;
- $m : V \to \mathcal{X}'$ is a surjective map between relevant variables in the base model $\mathcal{M}$ and variables in the abstracted model $\mathcal{M}'$;

- $\alpha_{X'} : \mathbb{D}[m^{-1}(X')] \to \mathbb{D}[X']$ is a collection of surjective maps from the outcome of base variables onto outcomes of the abstracted variables.

An abstraction maps variables and realizations in the base SCM $\mathcal{M}$ onto variables and realizations in the abstracted SCM $\mathcal{M}'$. Notice that interventions can be immediately transported by abstracting the intervened variables and values. For simplicity, whenever clear from the context, we shorthand the application of $\alpha$ as in Tab. 2. In order to assess whether the causal effect of interventions is maintained, the notion of interventional consistency is introduced:

**Definition 2.4** (Interventional consistency error [Rischel, 2020, Zennaro et al., 2023b]). Given an abstraction $\alpha$ from $\mathcal{M}$ to $\mathcal{M}'$, let $\mathcal{J}$ be a set of pairs $(\mathrm{do}(\mathbf{x}), \mathbf{Y})$ with $(\mathbf{X}, \mathbf{Y}) \subseteq \mathcal{X}^2$ and $\mathbf{X} \cap \mathbf{Y} = \emptyset$. The interventional consistency (IC) error $e(\alpha)$ is defined as the greatest distance between the two paths on the following diagram:

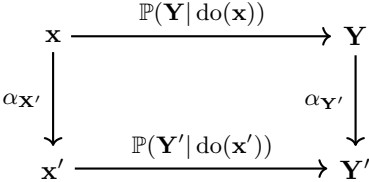

that is,

$$e(\alpha) = \max_{(\mathrm{do}(\mathbf{x}), \mathbf{Y}) \in \mathcal{J}} \mathrm{D_J}(\alpha(\mathbb{P}(\mathbf{Y} \mid \mathrm{do}(\mathbf{x}))), \mathbb{P}(\mathbf{Y}' \mid \alpha(\mathrm{do}(\mathbf{x}))))$$ (3)

where $\mathrm{D_J}(p, q)$ is the Jensen-Shannon (JS) distance between the distributions $p, q$ (see definition in App. A).

The IC error evaluates the worst-case difference between the distributions computed: (i) by intervening on $\mathcal{M}$ and then abstracting; or (ii) by abstracting and then intervening on $\mathcal{M}'$. An *exact abstraction* (w.r.t. the set $\mathcal{J}$) has $e(\alpha) = 0$, meaning that interventions and abstractions commute.

## 3 CAMABS

We are now ready to define the CAMAB problem:

**Definition 3.1.** A causally abstracted MAB (CAMAB) is defined by two CMABs, $\mathcal{B} = \langle \mathcal{M}, \mathcal{I} \rangle$ and $\mathcal{B}' = \langle \mathcal{M}', \mathcal{I}' \rangle$, and an abstraction $\alpha = \langle V, m, \alpha \rangle$ from $\mathcal{M}$ to $\mathcal{M}'$.

We will make the assumption that the models agree on the target, $m(Y) = Y'$, and that all relevant interventions in the base model can be mapped to the abstracted model.

*Example* 3.2. Let $\mathcal{B} = \langle \mathcal{M}, \mathcal{I} \rangle$ be a CMAB modelling a generic treatment-mediator-outcome SCM $\mathcal{M}$, as in Fig. 1(left), with $\mathcal{I} = \{\mathrm{do}(\mathtt{T} = 0), \mathrm{do}(\mathtt{T} = 1), \mathrm{do}(\mathtt{T} = 2)\}$

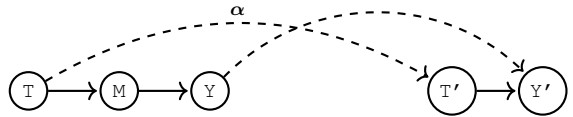

Figure 1: Base model $\mathcal{M}$ (left) and abstracted model $\mathcal{M}'$ (right)

encoding treatments with different dosages. Let $\mathcal{B}' = \langle \mathcal{M}', \mathcal{I}' \rangle$ be a simpler CMAB defined on a treatment-outcome SCM $\mathcal{M}'$, as in Fig. 1(right), with $\mathcal{I}' = \{\mathrm{do}(\mathtt{T}' = 0), \mathrm{do}(\mathtt{T}' = 1)\}$. A CAMAB is defined by an abstraction $\alpha = \langle V, m, \alpha \rangle$ between the two SCMs $\mathcal{M}$ and $\mathcal{M}'$, such as:

- $V = \{\mathtt{T}, \mathtt{Y}\}$;
- $m(\mathtt{T}) = \mathtt{T}', m(\mathtt{Y}) = \mathtt{Y}'$;
- $\alpha_{\mathtt{T}'}(0) = 0, \alpha_{\mathtt{T}'}(1) = 1, \alpha_{\mathtt{T}'}(2) = 1$, and $\alpha_{\mathtt{Y}'}$ is the identity map.

A CAMAB provides then a formal way to relate two CMABs defined on different variables with incompatible domains, and it thus offers a formulation to express the problem of transporting information across heterogeneous models. CMABs can then be related to each other through an abstraction, and their data can be aggregated to solve both CMABs jointly more efficiently.

## 4 MEASURING ERROR IN CAMABS

An abstraction in a CAMAB may introduce approximations. In order to estimate the discrepancy between the CMABs in a CAMAB, we rely on two measures. First, we introduce a variant of *IC error* to quantify the gap between the pushforward of the base model via $\alpha$ and the true abstracted model. As we focus on the reward variable $Y$, we reduce the set $\mathcal{J}$ in Def. 2.4 to the set $\mathcal{I}$ of pairs of the form $(\mathrm{do}(\mathbf{x}), Y)$ corresponding to relevant interventions. Moreover, we substitute the JS distance $\mathrm{D_J}$ with the Wasserstein 2-distance $\mathrm{D_{W_2}}$ (see definition in App. A); both distances have the 1-Lipschitz property that guarantees a bound on the error in the composition of abstractions [Rischel, 2020], but $\mathrm{D_{W_2}}(p, q)$ also allows us to bound the distance between the expected values of the distributions $p$ and $q$. We then redefine the IC error as:

$$e(\alpha) = \max_{\mathrm{do}(\mathbf{x}) \in \mathcal{I}} \mathrm{D_{W_2}}(\alpha(\mathbb{P}(Y \mid \mathrm{do}(\mathbf{x}))), \mathbb{P}(Y' \mid \alpha(\mathrm{do}(\mathbf{x})))).$$ (4)

This error quantifies the worst-case distributional distance between the reward distribution computed by (i) running action $\mathrm{do}(\mathbf{x})$ and pushing forward the resulting distribution via $\alpha_{Y'}$ in the abstracted CMAB, and (ii) abstracting the base action to $\alpha(\mathrm{do}(\mathbf{x}))$ and computing the reward distribution in the abstracted CMAB. A zero IC error in a CAMAB

Table 1: Equivalence between the CMAB Language and the SCM Formalism.

| | Action | Act. set | Reward | Rew. dom. | Rew. distr. | True exp. reward | Est. exp. reward |
|---|---|---|---|---|---|---|---|
| CMAB | $a_i$ | $\mathcal{A}$ | $g$ | $\mathbb{D}[G]$ | $G_{a_i}$ | $\mu_a = \mathbb{E}_{G_{a_i}}[G]$ | $\hat{\mu}_a = \hat{\mathbb{E}}_{G_{a_i}}[G]$ |
| Causal | $\mathrm{do}(\mathbf{x}_i)$ | $\mathcal{I}$ | $y$ | $\mathbb{D}[Y]$ | $\mathbb{P}(Y\,|\,\mathrm{do}(\mathbf{x}_i))$ | $\mu_{\mathrm{do}(\mathbf{x}_i)} = \mathbb{E}_{Y\,|\,\mathrm{do}(\mathbf{x}_i)}[Y]$ | $\hat{\mu}_{\mathrm{do}(\mathbf{x}_i)} = \hat{\mathbb{E}}_{Y\,|\,\mathrm{do}(\mathbf{x}_i)}[Y]$ |

Table 2: Shorthand Notation for the Application of an Abstraction to Variables, Values, Interventions, Distributions, and Abstracted Values.

| Shorthand | $\boldsymbol{\alpha}(X)$ | $\boldsymbol{\alpha}(x_i)$ | $\boldsymbol{\alpha}(\mathrm{do}(x_i))$ | $\boldsymbol{\alpha}(\mathbb{P}(Y))$ | $\boldsymbol{\alpha}^{-1}(x'_i)$ |
|---|---|---|---|---|---|
| Exact expression | $m(X)$ | $\alpha_{m(X'_i)}(x_i)$ | $\mathrm{do}(m(X_i) = \alpha_{m(X_i)}(x_i))$ | $(\alpha_{m(Y')}\#\mathbb{P})(Y')$ | $\{x_j \in \mathbb{D}[X] : \boldsymbol{\alpha}(x_j) = x'_i\}$ |

guarantees that the abstraction of the distribution of the outcome under the base action $a$ is the same as the distribution of the outcome under the abstracted action $\boldsymbol{\alpha}(a)$, that is, $\boldsymbol{\alpha}(\mathbb{P}(Y\,|\,\mathrm{do}(\mathbf{x}))) = \mathbb{P}(Y'|\boldsymbol{\alpha}(\mathrm{do}(\mathbf{x})))$.

We also introduce a second measure of *reward discrepancy* to quantify the gap between the reward distribution in the base model and the reward distribution in the pushforward of the base model:

$$s(\boldsymbol{\alpha}) = \max_{\mathrm{do}(\mathbf{x})\in\mathcal{I}} \mathrm{D}_{\mathrm{W}_2}(\mathbb{P}(Y\,|\,\mathrm{do}(\mathbf{x})), \boldsymbol{\alpha}(\mathbb{P}(Y\,|\,\mathrm{do}(\mathbf{x})))), \tag{5}$$

where the distributions are defined on $[0, 1]$. This error quantifies the worst-case distributional distance between the reward distribution computed by (i) running action $\mathrm{do}(\mathbf{x})$ in the base model, and (i) running the same action $\mathrm{do}(\mathbf{x})$ and pushing forward the resulting distribution via $\alpha_{Y'}$ in the abstracted CMAB. A *zero discrepancy* guarantees that the distribution of the rewards is identical in the base and under the pushforward.

These two measures immediately allow to bound the difference in expected values (see App. C for the formal proof):

**Proposition 4.1** (Bound on difference of expected rewards). *Given a CAMAB, the difference in expected rewards $|\mu_{a_i} - \mu'_{\boldsymbol{\alpha}(a_i)}|$ is bound by $e(\boldsymbol{\alpha}) + s(\boldsymbol{\alpha})$.*

# 5 TRANSFERRING INFORMATION IN CAMABS

In our general CAMAB formulation transfer may be characterized along multiple dimensions, such as:

- *Quantities to abstract:* whether transferring individual quantities (e.g.: actions, rewards) or model-based statistical quantities (e.g.: expectations).

- *Synchronicity of abstraction:* whether transferring information online (solving the CMABs synchronously) or offline (solving the CMABs asynchronously);

- *Direction of abstraction:* whether transferring information from base to abstracted model, or vice versa.

We consider a series of three representative scenarios in order of increasing complexity, covering meaningful approaches to the solution of a CAMAB. We focus, in particular, on the dimension of the *quantity to abstract*, assuming an offline transfer of information from the base to the abstracted model. For each scenario, we propose algorithmic solutions, analyze their behaviour in terms of regret, and provide examples; we refer to App. C for formal proofs, App. D for pseudo-code of our algorithms, App. E for details about our examples, and to the online repository[1] for the simulation code.

## 5.1 TRANSFER OF OPTIMAL ACTION

Let us consider a CAMAB where the base CMAB $\mathcal{B}$ was solved using a standard CMAB algorithm (see Alg. 4), so that at timestep $T$ an optimal action $a_o$ has been identified. As abstraction learning algorithms often work by minimizing the IC error [Zennaro et al., 2023a, Felekis et al., 2023], it may seem sensible that an exact abstraction would allow to solve the CMAB problem $\mathcal{B}'$ simply by transferring the optimal action. We could then consider the following intuitive protocol: we use the abstraction $\boldsymbol{\alpha}$ to transport the optimal action in $\mathcal{B}$ to $\mathcal{B}'$, and keep choosing action $\boldsymbol{\alpha}(a_o)$. This defines a new *transfer-optimum* (TOpt) algorithm, as in Alg. 1. This protocol is very efficient, requiring only one computation for transferring $a_o$. However, we now show a negative result:

**Proposition 5.1** (Biasedness of TOpt). *Assuming an exact abstraction $\boldsymbol{\alpha}$ and the optimality of the learned action $a_o = a^*$, it is not guaranteed that $\boldsymbol{\alpha}(a_o) = a'^*$.*

Although counterintuitive at first, this proposition can be proved through counterexamples as follows.

*Example* 5.2. Consider the CAMAB in Fig.1 where $\mathcal{M}$ is defined on binary variables. As long as the base composed mechanism $f_Y(f_M())$ is symmetric, and $f_{Y'}() = f_Y(f_M())$, then a zero IC error may be achieved by taking both $\alpha_{T'}$ and

---

[1] https://github.com/FMZennaro/causally-abstracted-multiarmed-bandits

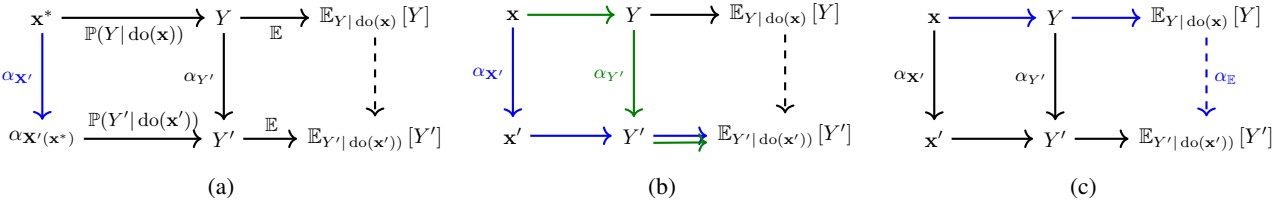

Figure 2: Diagrams illustrating the (a) `TOpt`, (b) `IMIT`, (c) `TExp` algorithms

$\alpha_{Y'}$ as anti-diagonal matrices. Then $e(\boldsymbol{\alpha}) = 0$, but $\boldsymbol{\alpha}(a^*) \neq a'^*$. See Fig. 4b in Appendix for a concrete illustration.

The symmetry in the mechanisms allows us to swap the labels of interventions and outcomes, while preserving consistency. This limit case may be unlikely in real scenarios, where causal asymmetries rule out such a possibility. Moreover, since in a MAB the domain $\mathbb{D}[Y]$ has an implicit ordering, we can prevent this by requiring $\alpha_{Y'}$ to be *order-preserving*. There is however another counterexample.

*Example* 5.3. Let $\mathcal{B}$ and $\mathcal{B}'$ be CMABs with the same underlying DAG in Fig. 1(right). Let us define an order-preserving zero-error abstraction $\boldsymbol{\alpha}$ between them. If the domains of the outcomes can be different $\mathbb{D}[Y] \neq \mathbb{D}[Y']$, we can be able to choose $\mathbb{D}[Y']$ so that $\sum_{y' \in \mathbb{D}[Y]} y' \boldsymbol{\alpha}(\mathbb{P}(Y|\mathrm{do}(a^*))) \leq \sum_{y' \in \mathbb{D}[Y]} y' \mathbb{P}(Y'|\mathrm{do}(a'^*)))$, thus implying that $\boldsymbol{\alpha}(a^*) \neq a'^*$. See Fig. 5 for a concrete illustration.

Even with an exact abstraction $\boldsymbol{\alpha}$ and an order-preserving map $\alpha_{Y'}$, the weighting of the interventional distributions by the different values in $\mathbb{D}[Y]$ and $\mathbb{D}[Y']$ can lead to different expected values. In general, these examples make sense as an exact abstraction guarantees that commuting interventions and abstraction produces the same distribution, but it does not guarantee that maxima in the base distribution will necessarily be mapped onto maxima in the abstracted model, that is:

$$\boldsymbol{\alpha}(\mathbb{P}(Y|\mathrm{do}(\mathbf{x}))) = \mathbb{P}(Y'|\boldsymbol{\alpha}(\mathrm{do}(\mathbf{x}))) \not\Rightarrow \quad (6)$$

$$\boldsymbol{\alpha}\left(\operatorname*{argmax}_{\mathrm{do}(\mathbf{x}) \in \mathcal{I}} \mu_{\mathrm{do}(\mathbf{x})}\right) = \operatorname*{argmax}_{\mathrm{do}(\mathbf{x}') \in \mathcal{I}'} \mu'_{\mathrm{do}(\mathbf{x}')} \quad (7)$$

Here, we are not interested in the commutativity of the diagram of Def. 2.4, but in the outer path of a diagram as in Fig. 2a; `TOpt` relies only on map $\alpha_{\mathbf{x}'}$ (blue arrow) improperly assuming the maximum of the expected values to be aligned (dashed arrow). However, the commutativity of the leftmost square encoded by Eq. 6 does not imply the commutativity of the outer square required by Eq. 7.

Prop. 4.1 allows us to define a sufficient condition under which the maximum is preserved, as requested by Eq. 7.

**Lemma 5.4** (Sufficient condition for preservation of maximum). *Given a CAMAB, the maximum is preserved by the abstraction $\boldsymbol{\alpha}$ is $e(\boldsymbol{\alpha}) + s(\boldsymbol{\alpha}) \leq \frac{1}{2} \min_{a \in \mathcal{A}: \Delta(a) > 0} \Delta(a)$.*

An alternative algebraic formulation of a sufficient condition is provided in Lemma C.1 in Appendix. Preservation of maxima immediately determines the asymptotic simple regret $\bar{R}'_{\text{TOpt}}(T)$ of the abstracted CMAB:

**Proposition 5.5** (Asymptotic regret of `TOpt`). *For $T \to \infty$, using* `TOpt`*, $\bar{R}'_{\text{TOpt}}(T) \to 0$ iff $\boldsymbol{\alpha}(a^*) = a'^*$.*

Although on a finite time the algorithm on the base CMAB may wrongly hold as optimal an action $a_o$ which happens to map to the optimal action, that is $\boldsymbol{\alpha}(a_o) = a'^*$, in the asymptotic regime maximum preservation is required to achieve zero simple regret in the abstracted CMAB.

*Example* 5.6. Let us consider two CAMABs as the one in Ex. 5.2, both with an exact abstraction, but not necessarily maximum-preserving. Fig. 3a shows that, in case of maximum preservation (blue), the simple regret of `UCB` (dashed) and `TOpt` (solid) both converges to 0 as a function of the timesteps $T$; instead, in case of no maximum preservation (green), the simple regret of `TOpt` (solid) does not converge to 0 as `UCB` (dashed). The results agree with the statement of Prop. 5.5.

Even if appealing, we have shown that `TOpt` is expected to perform satisfactorily only under the stringent condition that the maximum is preserved; otherwise, the abstracted CMAB will incur in a linear cumulative regret.

---

**Algorithm 1** `TOpt` Algorithm

---

1: **Input:** CMAB $\mathcal{B}$, estimated optimal action $a_o$ in the base model, abstraction $\boldsymbol{\alpha}$
2: **Output:** optimal policy $\pi$
3: Set $\pi(\boldsymbol{\alpha}(a_o)) = 1$
4: **Return:** $\pi$

---

## 5.2 TRANSFER OF ACTIONS

Instead of transferring the optimal action, we may consider an algorithm inspired by *off-policy reinforcement learning* (RL) where, at each timestep $t$, the abstracted agent translates the outcome of the decision-making of the base agent and takes action $\boldsymbol{\alpha}(a^{(t)})$. This process is illustrated by the blue arrows in Fig. 2b and codifies an *imitation* algorithm (`IMIT`), as in Alg. 2. In RL terms, the abstracted agent uses

---

**Algorithm 2** `IMIT` Algorithm

---

1: **Input:** CMAB $\mathcal{B}$, set of trajectories $\mathcal{D} = \{(a^{(t)}, g^{(t)})\}_{t=1}^{T}$ from base model, abstraction $\boldsymbol{\alpha}$
2: **Output:** optimal policy $\pi$
3: Initialize expected rewards $\hat{\mathbb{E}}_{Y|a_i}^{(0)}[Y]$, auxiliary statistics $\hat{\mathcal{S}}^{(0)}$ {Setup the params}
4: **for** $t = 1...T$ **do**
5:    Select $a^{(t)} \leftarrow \boldsymbol{\alpha}(\mathcal{D}[0, t])$ {Action-translation}
6:    Receive $g^{(t)} \sim G_{a^{(t)}}$ {Reward-collection}
7:    Compute $\hat{\mathbb{E}}_{Y|a^{(t)}}^{(t)}[Y], \hat{\mathcal{S}}^{(t)} \leftarrow$ update $\left(\hat{\mathbb{E}}_{Y|a^{(t)}}^{(t-1)}[Y], \hat{\mathcal{S}}^{(t-1)}, a^{(t)}, g^{(t)}\right)$ {Update stats}
8: **end for**
9: Compute $\pi^{(T)} \leftarrow$ `ALG` $\left(\hat{\mathbb{E}}_{Y|a^{(t)}}^{(T)}[Y], \hat{\mathcal{S}}^{(T)}\right)$ {Evaluate policy}
10: **Return:** $\pi^{(T)}$

---

the *behaviour policy* $\boldsymbol{\alpha}(\pi^{(t)})$ to train its *target policy* $\pi'^{(t)}$ [Sutton and Barto, 2018]. Unfortunately, the IC error itself is not informative regarding the distance between $\boldsymbol{\alpha}(\pi^{(t)})$ and $\pi'^{(t)}$, as the commutativity implied by zero IC error does not relate to the distribution of actions:

$$\boldsymbol{\alpha}(\mathbb{P}(Y|\operatorname{do}(\mathbf{x}))) = \mathbb{P}(Y'|\boldsymbol{\alpha}(\operatorname{do}(\mathbf{x}))) \not\Rightarrow \quad (8)$$
$$\mathbb{P}(\operatorname{do}(\mathbf{x}')) = \mathbb{P}(\boldsymbol{\alpha}(\operatorname{do}(\mathbf{x}))). \quad (9)$$

Still, as actions are run in the abstracted CMAB, it follows:

**Lemma 5.7** (Unbiasedness of `IMIT`). *Under the coverage assumption that $\pi^{(t)}$ has non-zero probability for every action $a^{(t)}$, the estimates $\hat{\mu}_{\operatorname{do}(\mathbf{x})}$ of `IMIT` are unbiased.*

Granted the coverage assumption, it holds that the expected rewards learned directly on the abstracted CMAB or learned on the abstracted CMAB by translating the actions of the base CMAB will be equal. However, CMAB algorithms will not always satisfy the coverage assumption; instead, algorithms like `UCB` will learn to choose the optimal action $a^*$. For a finite $T$, the confidence in the estimate of the expected rewards will depend on the number of times each action $a'$ is tested under `IMIT`. Let $\mathcal{K}(a_i') = |\boldsymbol{\alpha}^{-1}(a_i')|$ be the size of the set of base actions mapping to $a_i'$; then, we can derive the following result in relation to `UCB`:

**Proposition 5.8** (Confidence of `IMIT`). *Given a CAMAB, assume we have run `UCB` for $T$ steps on $\mathcal{M}$. For `IMIT` to reach the same level of confidence in $\hat{\mu}_{a_i'}$ as running `UCB` for $T$ steps on $\mathcal{M}'$, it must hold $N(\mathcal{K}(a_i') - 1) + \left(\sum_{a_j \in \boldsymbol{\alpha}^{-1}(a_i')} \frac{1}{\Delta(a_j)^2} - \frac{1}{\Delta(a_i')^2}\right) \geq 0$ where $N > 0$ is a constant term.*

The first term of the inequality accounts for a constant number of time each action has to be sampled; in the abstracted CMAB, an action $a_i'$ aggregates the constant component from all the base actions in $\boldsymbol{\alpha}^{-1}(a_i')$. The second term of the inequality accounts for an additional number of times each action has to be sampled according to its optimality gap $\Delta(a_i')$; again, in the abstracted CMAB, an action $a_i'$ again aggregates the gap from all the base actions $a_j \in \boldsymbol{\alpha}^{-1}(a_i')$.

A similar reasoning can be followed to discuss the regret when running `IMIT` against running another algorithm directly on the abstracted CMAB:

**Lemma 5.9** (Cumulative regret for `IMIT`). *Given a CAMAB, assume we have run `ALG` for $T$ steps on $\mathcal{M}$. The difference in cumulative regret between running `ALG` or `IMIT` on $\mathcal{M}'$ is $\mathbb{E}_{\text{ALG}}\left[\sum_{t=0}^{T} \sum_{a_i' \in \mathcal{A}'} \boldsymbol{\alpha}(\mathbb{P}(a'^{(t)} = a_i'))\mu_{a_i'}\right] - \mathbb{E}_{\text{ALG}}\left[\sum_{t=0}^{T} \sum_{a_i' \in \mathcal{A}'} \mathbb{P}(a'^{(t)} = a_i')\mu_{a_i'}\right].$*

The difference in cumulative regret is dependent on the weighting of the policies $\boldsymbol{\alpha}(\pi)$ and $\pi'$. More concretely, if we take `ALG` to be `UCB` we obtain:

**Proposition 5.10** (Regret lower bound of `IMIT`). *Given a CAMAB, assume we have run `UCB` for $T$ steps on $\mathcal{M}$. For `IMIT` to have a lower regret bound than running `UCB` for $T$ steps on $\mathcal{M}'$, it must hold $3\sum_{a_i' \in \mathcal{A}'} \Delta(a_i')[1 - \mathcal{K}(a_i')] + 16\log T \sum_{a_i' \in \mathcal{A}'} \left[\frac{1}{\Delta(a_i')} - \Delta(a_i')\frac{1}{\sum_{\boldsymbol{\alpha}^{-1}(a_i')}\Delta^2(a_j)}\right] \geq 0.$*

Notice how the confidence in Prop. 5.8 and the lower bound in Prop. 5.10 are in a trade-off: if many actions $a_j$ with small gaps $\Delta(a_j)$ map onto $a_i'$, then `IMIT` will oversample $a_i'$ and be overconfident in its estimation; however, because of such an overestimation, its cumulative regret will be greater than just running `UCB`. `IMIT` thus performs best when the optimal action $a^*$ together with other actions $a_j$ with small gaps $\Delta(a_j)$ are mapped onto $a_i'^*$, as the oversampling will not factor in the regret. This agrees with the intuition of a good abstraction clustering together the optimal action with actions providing a close reward. This dynamic is confirmed in the asymptotic regime:

**Proposition 5.11** (Asymptotic regret for `IMIT`). *For $T \rightarrow \infty$, the abstracted CMAB using `IMIT` achieves sub-linear cumulative regret iff $\boldsymbol{\alpha}(a^*) = a'^*$.*

*Example* 5.12. Consider two CAMABs defined on the same CMABs, the first one with a non-maximum-preserving abstraction $\boldsymbol{\alpha}_1$ and the second with an abstraction $\boldsymbol{\alpha}_2$ that aggregates the maximum and another slightly suboptimal

**Algorithm 3** `TExp` Algorithm

---

1: **Input:** CMAB $\mathcal{B}$, estimate rewards $\hat{\mathbb{E}}^{(t)}_{Y|a_i}[Y]$ from the base model at timestep $t$, abstraction $\boldsymbol{\alpha}$, time horizon $T$
2: **Output:** optimal policy $\pi$
3: Initialize expected rewards $\hat{\mathbb{E}}^{(0)}_{Y'|a'_i}[Y']$ by abstracting $\hat{\mathbb{E}}^{(t)}_{Y|a_i}[Y]$ {Expected value-translation}
4: Reduce the action set $\mathcal{I}'$ to the set of optimistic interventions $\mathcal{I}'_+$ {Action sub-selection}
5: Initialize auxiliary statistics $\mathcal{S}^{(0)}$ and policy $\pi^{(0)}$ {Setup the params}
6: **for** $t = 1...T$ **do**
7:      Select $a'^{(t)} \sim \pi^{(t-1)}$ {Decision-making}
8:      Receive $g^{(t)} \sim G_{a'^{(t)}}$ {Reward-collection}
9:      Compute $\hat{\mathbb{E}}^{(t)}_{Y'|a'^{(t)}}[Y'], \hat{\mathcal{S}}^{(t)} \leftarrow \text{update}\left(\hat{\mathbb{E}}^{(t-1)}_{Y'|a'^{(t)}}[Y'], \hat{\mathcal{S}}^{(t-1)}, a'^{(t)}, g^{(t)}\right)$ {Update stats}
10:      Compute $\pi^{(t)} \leftarrow \text{ALG}\left(\hat{\mathbb{E}}^{(t)}_{Y'|a'^{(t)}}[Y'], \hat{\mathcal{S}}^{(t)}\right)$ {Update policy}
11: **end for**
12: **Return:** $\pi^{(T)}$

---

actions; see illustration in Fig. 8b and 9a, respectively. Fig. 3b shows that the difference between the regret of running `UCB` directly on the abstracted model and `IMIT` can be either negative (for $\boldsymbol{\alpha}_1$ in blue) or positive (for $\boldsymbol{\alpha}_1$ in red), as explained by Prop. 5.10.

On finite-time, `IMIT` has less strict conditions for success than `TOpt`; however, this assessment is not trivial, and `IMIT` still requires running the abstracted CMAB.

## 5.3 TRANSFER OF EXPECTED VALUES

We now consider the possibility of transferring the expected value of the rewards in the base CMAB in order to warm-start the abstracted CMAB. This approach corresponds to the computation represented by the blue arrows in Fig. 2c. Formally, if the outer square of Fig. 2c were to commute, then it would hold that:

$$\alpha_{\mathbb{E}}(\mu_{\text{do}(\mathbf{x})}) = \mu'_{\boldsymbol{\alpha}(\text{do}(\mathbf{x}))}, \tag{10}$$

guaranteeing that the abstraction of the expected rewards is the same as the expected rewards in the abstracted model. In order to define a map $\alpha_{\mathbb{E}}$, we extend $\alpha_{Y'} : \mathbb{D}[Y] \rightarrow \mathbb{D}[Y']$ to $\mathbb{R} \rightarrow \mathbb{R}$ by selecting from a function class $\mathcal{F}$ a map $\alpha_{\mathbb{E}}$ that interpolates the set of pairings $\mathcal{D} = \{(y, \alpha_{Y'}(y))\}$, for $y \in \mathbb{D}[Y]$. We then initialize each expected reward $\hat{\mu}'_{\text{do}(\mathbf{x}'_i)}$ in the abstracted CMAB by optimistically transferring the highest expected reward among the action mapping to $\mathbf{x}'_i$:

$$\hat{\mu}'_{\text{do}(\mathbf{x}'_i)} = \max_{\mathbf{x}_j \in \boldsymbol{\alpha}^{-1}(\mathbf{x}'_i)} \alpha_{\mathbb{E}}(\hat{\mu}_{\text{do}(\mathbf{x}_j)}) \tag{11}$$

With reference to Fig. 2c, we can now derive a bound on the difference between the upper path $\alpha_{\mathbb{E}}(\mu_{\text{do}(\mathbf{x})})$ denoting our approach, and the lower path $\mu'_{\boldsymbol{\alpha}(\text{do}(\mathbf{x}))}$ denoting a standard CMAB algorithm:

**Proposition 5.13** (Bias of $\alpha_{\mathbb{E}}$). *Assuming a linear interpolating function $\alpha_{\mathbb{E}}$, the difference $|\alpha_{\mathbb{E}}(\mu_{\text{do}(\mathbf{x})}) - \mu'_{\boldsymbol{\alpha}(\text{do}(\mathbf{x}))}|$*

*is upper bounded by $|\mathbb{E}_{Y|\text{do}(\mathbf{x})}[\epsilon_{Y'}(Y)]| + e(\boldsymbol{\alpha})$, where $\epsilon_{Y'}(Y)$ is the interpolation error introduced by $\alpha_{\mathbb{E}}$.*

Thus, the quality of the transfer of the expected reward is a function of the interpolation $\alpha_{\mathbb{E}}$ and the IC error $e(\boldsymbol{\alpha})$. Given an exact abstraction and a perfect linear interpolation, expected rewards in the base model can be exactly transported to the abstracted model. This bound allows us to define a confidence bound on the estimates set via $\alpha_{\mathbb{E}}$:

**Lemma 5.14** (Confidence bounds for $\alpha_{\mathbb{E}}$). *Assuming $\mathbb{D}[Y] = \mathbb{D}[Y'] = [0, 1]$ and assuming we used a linear interpolating function $\alpha_{\mathbb{E}}$ to compute $\hat{\mu}'_{\text{do}(\mathbf{x}'_i)}$ as in Eq. 11, with probability at least $1 - \delta$, it holds that $|\mu'_{\text{do}(\mathbf{x}'_i)} - \hat{\mu}'_{\text{do}(\mathbf{x}'_i)}| \leq \kappa$, where $\kappa = \sqrt{\frac{2\log(2/\delta)}{\mathcal{C}(\text{do}(a'_i))}} + |\mathbb{E}_{Y|\text{do}(\mathbf{x})}[\epsilon_{Y'}(Y)]| + e(\boldsymbol{\alpha})$ and $\mathcal{C}(\text{do}(\mathbf{x}'_i))$ counts the number of times action $\text{do}(\mathbf{x}'_i)$ was taken.*

We can then use this bound to restrict the action in the abstracted CMAB to a reduced set $\mathcal{I}'_+$ of optimistic interventions, that is, interventions $\text{do}(\mathbf{x}'_i)$ such that $\exists\, \text{do}(\mathbf{x}'_j) \in \mathcal{I}'$:

$$\hat{\mu}'_{\text{do}(\mathbf{x}'_i)} + \kappa \geq \hat{\mu}'_{\text{do}(\mathbf{x}'_j)} - \kappa. \tag{12}$$

We refer the algorithm that transfers the expected values via $\alpha_{\mathbb{E}}$, restricting the action set to $\mathcal{I}'_+$, and the runs `UCB` as `TExp` (see Alg. 3). We then immediately get:

**Proposition 5.15** (Cumulative regret of `TExp`). *Given a CAMAB, assume we use `TExp` to initialize the expected rewards of the abstracted CMAB, and then we run `UCB` for $T$ steps. Then the cumulative regret we incur is bounded by: $R'_{\text{TExp}}(T) \leq 3\sum_{a'_i \in \mathcal{I}'_+} \Delta(a'_i) + \sum_{a'_i \in \mathcal{I}'_+ : \Delta(a'_i) > 0} \frac{16 + \log(T)}{\Delta(a'_i)}$*

Notice that `TExp` relies on explicit knowledge of $\kappa$ which might not be available; in such case, the results on biasedness will still hold, but we would not be able to restrict

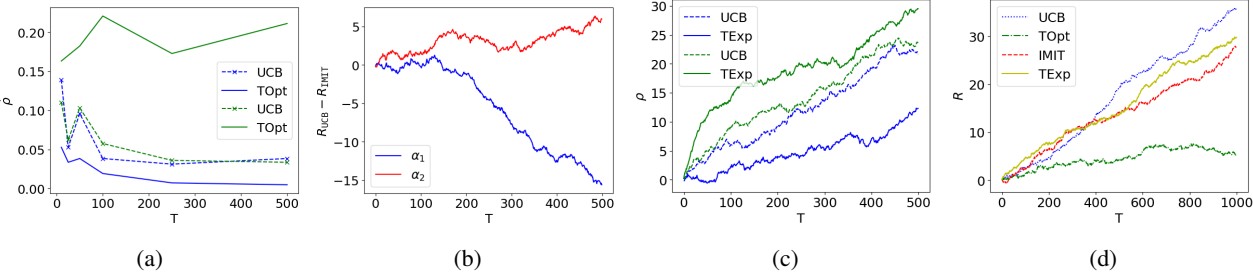

Figure 3: (a) Simple regret for `TOpt` from Ex. 5.6 for an exact and maximum-preserving abstraction (blue lines) and an exact but not maximum-preserving abstraction (green lines). (b) Regret difference for `IMIT` from Ex. 5.12 using abstractions aggregating values differently (red and blue lines). (c) Cumulative regret for `TExp` from Ex. 5.16 for an abstraction preserving domains (blue lines) and an abstraction changing domains (green lines). See respective examples for further explanation. (d) Cumulative regret on the online advertising scenario.

actions to $\mathcal{I}'_+$. We leave the problem of learning abstractions and estimating IC error to future work.

*Example* 5.16. Let us return a last time to the CAMAB of Ex. 5.2, and let us modify the outcome Y to be defined over $\mathbb{D}[Y] = \{0, 1, 2\}$; thanks to the perfect interpolation and the low IC error, Fig. 3c shows that `TExp` (blue solid) allows for a lower regret compared to running `UCB` on the abstracted CMAB (blue dashed); however, if we were to change the domain of Y to $\mathbb{D}[Y] = \{0.4, 0.5, 10\}$, thus inducing a higher IC and interpolation error $\alpha_{\mathbb{E}}$, then, as expected from Prop. 5.13, `TExp` (green solid) incurs in a higher regret than `UCB` (green dashed).

`TExp` provides an algorithm to compute and transfer bounds within a CAMAB. As a further extension, in App. D.1 we also present an method based on the transfer of individual rewards which joins the advantages of transferring individual quantities like `IMIT` and the algorithmic approach of `TExp`.

## 6 EXPERIMENTS

In this section, we present more experimental results aimed at (i) discussing the relation between our work and methods from the transfer literature, (ii) showcasing our algorithms on more realistic CMAB problems. Detailed results are provided in Appendix E.

Our CAMAB approach has a close resemblance to the `B-UCB` algorithm for transfer learning proposed in Zhang and Bareinboim [2017]. `TExp` can be seen as a generalization of `B-UCB`, based on the idea of transporting bounds between related CMABs; indeed, while `B-UCB` requires source and target SCMs to be defined on identical variables, our `TExp` can relate SCMs defined on different variables. In appendix, we show an application of our methods to some of the specific scenarios investigated in Zhang and Bareinboim [2017] by recasting them in terms of abstractions.

Finally, we compare our algorithms on a more realistic

CAMAB by reconsidering the online advertisement problem presented in the introduction. To define our base CMAB, we use the model presented in Lu et al. [2020a] based on data from Adobe (see Fig. 13). For our abstracted CMAB, we design a smaller model that could have been drawn up by another department; specifically, we assume that the new model simplifies the purposes and products of the campaign and ignores the advertisement sending out time. Notice that in this scenario, `B-UCB` would not be applicable, as the two models in the CAMAB are defined over different variables. Fig. 3d shows the cumulative regret of the proposed algorithms, highlighting how our CAMAB algorithms improve over directly running `UCB` on the abstracted model - all the results are further explored in terms of the chosen abstraction in Appendix.

## 7 RELATED WORK

The stochastic MAB problem has been studied extensively within the MAB literature; see Lattimore and Szepesvári [2020] for a thorough overview. In this setting, it is typically assumed that the reward distributions associated with each action are statistically independent. Variations of popular index-based algorithms, including `UCB`, become suboptimal once structural relationships between reward distributions are assumed [Russo and Van Roy, 2014, Lattimore and Szepesvari, 2017].

To address this, various structural MABs have been proposed which consider different kinds of statistical dependence between reward distributions, such as linear reward structures [Lattimore and Szepesvari, 2017, Rusmevichientong and Tsitsiklis, 2010, Valko et al., 2013, Li et al., 2019] and their generalisation to nonstationary settings [Russac et al., 2019, Zhao et al., 2020]. Similarly, no-regret learning algorithms have been designed for Lipschitz MABs [Magureanu et al., 2014, Kleinberg et al., 2019], wherein similar actions have similar expected reward. For finite-arm settings, Lazaric et al. [2013] and Lattimore and Munos

[2014] propose a `UCB`-style algorithm for a structured MAB where the joint reward distribution belongs to a finite feasible set or to a parametrised family, respectively. Inspired by Combes et al. [2017], Van Parys and Golrezaei [2023] study a general setting wherein the reward distribution belongs to a known convex set.

The use of causal models to encode structural relationships in MABs was first proposed by Bareinboim et al. [2015] in the form of MABs with unobserved confounders (MABUC). The CMAB problem, in which actions are interventions on an SCM with known causal graph, was formally introduced by Lattimore et al. [2016], who designed an algorithm with sublinear simple regret guarantees. Since then, various Lu et al. [2020b] `UCB`-style algorithms for CMABs have been proposed [Nair et al., 2021, Bilodeau et al., 2022, Lu et al., 2020b].

Lazaric [2012], Van Parys and Golrezaei [2023] have studied the problem of transferring information across MABs, but only recently this problem has been extended to CMABs. Transfer within the same model, but across different observational and interventional regimes, has long been explained by standard do-calculus [Tian and Pearl, 2002], and extended by Forney et al. [2017] to the counterfactual regime. Zhang and Bareinboim [2017], instead, relied on bounds to transfer information between CMABs defined on identical variables; our work tackles a similar problem while relying on CA to overcome the limitation of having the SCM defined over the same variables. Transfer of information between SCMs has also been modelled via generalized linear models [Feng and Chen, 2022] and causal Bayesian optimization [Aglietti et al., 2020a].

Causal abstraction was first formalized by Rubenstein et al. [2017] and Beckers and Halpern [2019] as a relation between the interventional distributions of SCMs at different levels of abstraction. In this work, we follow the formulation in Rischel [2020], which explicitly defines a mapping between the variables of models related by an abstraction. Further related work is discussed in App. F.

# 8 CONCLUSION

In this paper we have considered how CMAB problems at different levels of resolution could be related via a formal abstraction map, and how such a map could be used to transfer information and improve learning. We have formulated the CAMAB problem, defined relevant measures, and proposed a simple taxonomy of CAMABs. We have then studied some representative scenarios, providing algorithms and theoretical analysis. Specifically: (i) In the first scenario (`TOpt`) we showed that, for exact and non-exact abstractions alike, if we transfer the optimal action from the base CMAB we are not guaranteed that $\boldsymbol{\alpha}(a^*) = a'^*$ (Prop. 5.1); we discussed a sufficient condition for the preserva-

tion of the optimum (Lem. 5.4) and showed that, without preservation of the optimum, the abstracted CMAB using `TOpt` incurs in a constant simple regret (Prop. 5.5). (ii) In the second scenario (`IMIT`) we showed that running the abstracted CMAB using an exact or non-exact abstraction to transport actions, the confidence and the regret in the abstracted CMAB are in a trade-off (Prop. 5.8 and Prop. 5.10), which highlighted situations when `IMIT` may be expected to perform better than `UCB`; moreover, to reach an asymptotic sub-linear cumulative regret `IMIT` still needs optimum preservation (Prop. 5.11). (iii) Finally, in the third scenario (`TExp`), we showed that we can extend the abstraction as $\alpha_{\mathbb{E}}$ and directly transport expected rewards; we proved that the bias introduced by abstracting is bounded by the IC and the interpolation error (Prop. 5.13), and used this bound to define new confidence intervals (Lem. 5.14) and derive the cumulative regret of `TExp` (Prop. 5.15).

The theoretical and empirical results we have presented characterize the advantages of using an abstraction map, as well as its limitations. A naive use of an abstraction map may lead to sub-optimal results. In `TOpt`, counterintuitively, even an exact abstraction was no guarantee of a useful transfer. Indeed, we have shown how transporting information between CMABs may depend on non-obvious details of the abstraction itself. In the case of `IMIT`, the confidence and the regret of the abstracted CAMAB were in a trade-off depending on how the abstraction aggregates actions and modified their optimality gaps; in the case of `TExp` both IC error and interpolation error contributes to the quality of the final result. These non-trivial results highlight both the versatility of abstraction and the need for proper measures to evaluate its impact on a decision-making problem.

Summarising, our work extends the scope of transfer learning across CMABs by laying a theoretical foundation (CAMAB) for transferring and exploiting information across decision-making problems observed at multiple levels of resolution via causal abstraction. Potential directions for future work include the construction of efficient abstraction maps for CAMABs with well-aligned rewards, and the relationship between CAMABs and other specialised frameworks for MABs, such as regional and structured bandits.

**Acknowledgements**

NB, JD, AC, and MW acknowledge funding from a UKRI AI World Leading Researcher Fellowship awarded to Wooldridge (grant EP/W002949/1). MW and AC also acknowledge funding from Trustworthy AI - Integrating Learning, Optimisation and Reasoning (TAILOR), a project funded by European Union Horizon2020 research and innovation program under Grant Agreement 952215. YF: This scientific paper was supported by the Onassis Foundation - Scholarship ID: F ZR 063-1/2021-2022. TD acknowledges support from a UKRI Turing AI acceleration Fellowship

[EP/V02678X/1].

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

# Causally Abstracted Multi-armed Bandits
## (Supplementary Material)

**Fabio Massimo Zennaro**[1]  **Nicholas Bishop**[2]  **Joel Dyer**[2]  **Yorgos Felekis**[3]  **Anisoara Calinescu**[2]  **Michael Wooldridge**[2]  **Theodoros Damoulas**[3]

[1]University of Bergen
[2]University of Oxford
[3]University of Warwick

## A  ADDITIONAL DEFINITIONS

In this section we provide additional definitions.

**Definition A.1** (Jensen-Shannon Distance (JSD))**.** Let $p$ and $q$ be two distributions strictly positive on the domain $\mathbb{D}[X]$. The Jensen-Shannon distance is defined as:

$$\mathrm{D_J}(p, q) = \sqrt{\frac{1}{2}d_{KL}\left(p; \frac{p+q}{2}\right) + \frac{1}{2}d_{KL}\left(q; \frac{p+q}{2}\right)}$$

where:

$$d_{KL}(p, q) = -\sum_{x \in \mathcal{X}} p(x) \log \frac{p(x)}{q(x)}$$

is the Kullback-Leibler (KL) divergence.

**Definition A.2** (Wasserstein P-distance)**.** Let $p(x)$ and $q(x')$ be two distributions on the domain $\mathbb{D}[X]$ with finite $P$ moments, the Wasserstein P-distance is defined as:

$$\mathrm{D_{W_P}}(p, q) = \inf_{\pi \in \Pi(p,q)} \left(\mathbb{E}_\pi \left[d(x, x')^P\right]\right)^{\frac{1}{P}}$$

where $\pi$ is a joint distribution over $\mathbb{D}[X] \times \mathbb{D}[X]$ with marginals $p$ and $q$ from the set of all such possible joints $\Pi$, and $d(x, x')$ is a metric over $\mathbb{D}[X] \times \mathbb{D}[X]$.

We take our $\mathrm{D_{W_2}}(p, q)$ for $P = 2$, $\mathbb{D}[X] = \mathbb{R}$ or $\mathbb{D}[X] = [0, 1]$, and $d(x, x')$ as the Euclidean distance.

## B  ASSUMPTIONS

In this section we provide a complete listing of all our assumptions together with their explanation.

### B.1  SCM ASSUMPTIONS

Our SCM aligns with the standard assumptions proposed by Rischel [2020]:

(SCM1) *finite set of variables*: the SCM $\mathcal{M}$ is defined on a finite set $\mathcal{X}$ of endogenous variables.

(SCM2) *finite domains for the variables*: each endogenous variable $X \in \mathcal{X}$ is defined on a finite domain $\mathbb{D}[X]$. Notice that this assumption allows us to encode the structural function $f_i$ of a SCM $\mathcal{M}$ as stochastic matrices.

(SCM3) *acyclicity*: the DAG $\mathcal{G}_\mathcal{M}$ entailed by $\mathcal{M}$ is acyclic.

## B.2 MAB ASSUMPTIONS

Our definition of MAB subscribes to a set of common assumptions in the literature:

(MAB1) *independent reward distributions*: each reward distribution $G_{a_i}$ is independent from other random variables.

(MAB2) *stationary reward distributions*: reward distributions $G_{a_i}$ do not change in time.

(MAB3) *no context*: no contextual information is provided to the agent before taking the action.

(MAB4) *finite variance of the distributions*: $\mathbb{V}_{G_{a_i}}[G] < \infty$, for all $a_i$. A stronger assumption is:

(MAB4+) *Bernoulli reward distributions*: $G_{a_i} = \texttt{Bernoulli}(p_i)$ for all $a_i$, where $\texttt{Bernoulli}(p)$ is a Bernoulli random variable with parameter $p$.

## B.3 CMAB ASSUMPTIONS

Our CMAB inherits some of the MAB assumptions (CMAB2-CMAB4), as well as introducing new ones (CMAB5-CMAB7):

(CMAB2) *stationary reward process*: the outcome of the target variable $Y$ does not change in time; this assumption is implied by the stationarity of the mechanisms of the SCM.

(CMAB3) *no context*: no observational data is provided before action.

(CMAB4) *finite variance of the process*: $\mathbb{V}_{Y|\operatorname{do}(\mathbf{x}_i)}[Y] < \infty$, for all $\mathbf{x}_i \in \mathcal{I}$.

(CMAB5) *limited set of intervenable variables*: only a subset of endogenous variables may be intervened upon [Lee and Bareinboim, 2019, Lu et al., 2020a, Aglietti et al., 2020b].

(CMAB6) *known DAG*: the DAG underlying the SCM is given.

(CMAB7) *finite domain of the rewards*: reward values $g$ lie in the interval $[0, 1]$ almost surely.

## B.4 ABSTRACTION ASSUMPTIONS

We consider abstractions that comply with the following assumptions:

(AB1) *partially specified abstraction:* we will assume that the abstraction $\boldsymbol{\alpha}$ is specified w.r.t. to the relevant variables for the CMAB problem, but not necessarily completely specified. This assumption decomposes as follows:

(AB1a) *base interventions are on relevant variables:* for each action $a_i$ corresponding to intervention $\operatorname{do}(\mathbf{X}_i = \mathbf{x}_i)$, then $\mathbf{X}_i \subseteq V$. Violation of this assumption would mean that certain actions in the base CMAB do not have any equivalent in the abstracted CMAB.

(AB1b) *base interventions have an image:* for each action $a_i$ corresponding to intervention $\operatorname{do}(\mathbf{X}_i = \mathbf{x}_i)$, then $m(\mathbf{X}_i) = \mathbf{X}'_j$, $\alpha_{\mathbf{X}'_j}(\mathbf{x}_i) = \mathbf{x}'_j$, and $\operatorname{do}(\mathbf{X}'_j = \mathbf{x}'_j) = a'_j \in \mathcal{A}'$. Violation of this assumption would mean that certain actions in the base CMAB can not properly be mapped onto actions in the abstracted CMAB.

(AB1c) *abstracted interventions have a counterimage:* For each intervention $a'_j \in \mathcal{A}'$ there is a corresponding intervention $a_i \in \mathcal{A}$ mapping onto it via abstraction $\boldsymbol{\alpha}$. This assumption implies that the simplified CMAB $\mathcal{B}'$ does not introduce new actions not available in the base CMAB $\mathcal{B}$.

If the abstraction were not known, then this could lead us to consider a CAMAB problem where we want to learn at the same time an optimal policy and an abstraction map.

(AB2) *abstraction not necessarily exact:* we will not assume the abstraction to be exact.

(AB3) *agreement on target:* we will assume $m(Y) = Y'$, that is, the two models have unique target variables, which are mapped to each other; resolution on the variable can change. This assumption implies that no factors are conflated in the target variable in the high-level model.

# C PROOFS

In this section we provide proofs for the lemmata and propositions in the paper.

## C.1 PROOF OF PROPOSITION 4.1

**Proposition 4.1** (Bound on difference of expected rewards). *Given a CAMAB, the difference in expected rewards* $|\mu_{a_i} - \mu'_{\boldsymbol{\alpha}(a_i)}|$ *is bound by* $e(\boldsymbol{\alpha}) + s(\boldsymbol{\alpha})$.

*Proof.* Let us consider a CAMAB defined on $\mathcal{M}$ and $\mathcal{M}'$. The IC error $e(\boldsymbol{\alpha})$ evaluates the worst-case distance over possible interventions between the reward distribution in the pushforward of the base model and the abstracted model as:

$$e(\boldsymbol{\alpha}) = \max_{\mathrm{do}(\mathbf{x}) \in \mathcal{I}} \mathrm{D}_{\mathrm{W}_2}(\boldsymbol{\alpha}(\mathbb{P}(Y \,|\, \mathrm{do}(\mathbf{x}))), \mathbb{P}(Y' | \boldsymbol{\alpha}(\mathrm{do}(\mathbf{x})))).$$

Because of the Wasserstein 2-distance this immediately provide also a bound on the worst case distance between the expected value in the pushforward of the base model and the abstracted model as:

$$|\boldsymbol{\alpha}(\mu_{\mathrm{do}(\mathbf{x})}) - \mu'_{\boldsymbol{\alpha}(\mathrm{do}(\mathbf{x}))}| \leq e(\boldsymbol{\alpha}).$$

Furthermore, the reward discrepancy quantifies the worst-case distance over possible interventions between the reward distribution in the base model and in the pushforward of the base model as:

$$s(\boldsymbol{\alpha}) = \max_{\mathrm{do}(\mathbf{x}) \in \mathcal{I}} \mathrm{D}_{\mathrm{W}_2}(\mathbb{P}(Y \,|\, \mathrm{do}(\mathbf{x})), \boldsymbol{\alpha}(\mathbb{P}(Y \,|\, \mathrm{do}(\mathbf{x})))),$$

Again, because of the Wasserstein 2-distance this immediately provide also a bound on the worst case distance between the expected value in the base model and the pushforward of the base model:

$$|\mu_{\mathrm{do}(\mathbf{x})} - \boldsymbol{\alpha}(\mu_{\mathrm{do}(\mathbf{x})})| \leq s(\boldsymbol{\alpha}).$$

By the triangular inequality, we get:

$$|\mu_{\mathrm{do}(\mathbf{x})} - \mu'_{\boldsymbol{\alpha}(\mathrm{do}(\mathbf{x}))}| \leq e(\boldsymbol{\alpha}) + s(\boldsymbol{\alpha}).$$

That is, $|\mu_{a_i} - \mu'_{\boldsymbol{\alpha}(a_i)}| \leq e(\boldsymbol{\alpha}) + s(\boldsymbol{\alpha})$. ∎

## C.2 PROOF OF PROPOSITION 5.1

**Proposition 5.1** (Biasedness of `TOpt`). *Assuming an exact abstraction* $\boldsymbol{\alpha}$ *and the optimality of the learned action* $a_o = a^*$, *it is not guaranteed that* $\boldsymbol{\alpha}(a_o) = a'^*$.

*Proof.* We prove this proposition through two counterexamples.

*First counterexample.* In the first counterexample, we consider the setup in Fig. 4a and 4b where intrinsic symmetries allow for multiple exact abstractions. Notice that all variables are binary.

Let us first consider the case in Fig. 4a.

The base SCM $\mathcal{M}$ is defined on binary variables and has structural functions given by the following matrices $f_T = \begin{bmatrix} .8 \\ .2 \end{bmatrix}$,

$f_M = \begin{bmatrix} .2 & .8 \\ .8 & .2 \end{bmatrix}, f_Y = \begin{bmatrix} .7 & .3 \\ .3 & .7 \end{bmatrix}$. The abstracted model $\mathcal{M}'$ is also defined on binary variables and has the following structural functions $f_{T'} = f_T$ and $f_{Y'} = f_Y f_M$ as a matrix product. Let $\boldsymbol{\alpha}$ be defined on $V, m$ as in Example 3.2, and let $\alpha_{T'}, \alpha_{Y'}$ being identity matrices. Fig. 4a summarizes the setup. Assuming $\mathcal{I} = \{\mathrm{do}(T = 0), \mathrm{do}(T = 1)\}$, the IC error for this abstraction $\boldsymbol{\alpha}$ is $e(\boldsymbol{\alpha}) = 0$. Moreover, the true expected rewards in the base CMAB are $\mathbb{E}_{Y | \mathrm{do}(T=0)}[Y] = 0.62$ and $\mathbb{E}_{Y | \mathrm{do}(T=1)}[Y] = 0.38$, leading to the optimal action $a^*$ being $\mathrm{do}(T = 0)$. Identically, in the abstracted CMAB we have $\mathbb{E}_{Y' | \mathrm{do}(T'=0)}[Y'] = 0.62$ and $\mathbb{E}_{Y' | \mathrm{do}(T'=1)}[Y'] = 0.38$, similarly leading to the optimal action $a'^*$ being $\mathrm{do}(T' = 0)$. Thus, if we use the abstraction $\boldsymbol{\alpha}$ to translate the optimal action $a^*$ we would map it to $\boldsymbol{\alpha}(a^*) = \mathrm{do}(T' = 0)$. Therefore, $a'^* = \boldsymbol{\alpha}(a^*)$.

Let us now consider the alternative abstraction in Fig. 4b, where domains, mechanisms and abstraction are exactly as above, except for $\alpha_{T'}, \alpha_{Y'}$ which are now anti-diagonal matrices. See Fig. 4b for an illustration of this setup. With respect to the same intervention set $\mathcal{I}$, it still holds that $e(\boldsymbol{\alpha}) = 0$. Furthermore all the expected rewards and the optimal actions $a^*, a'^*$ are similarly unchanged. However, the translation of $a^*$ via $\boldsymbol{\alpha}$ is now reversed, leading to $\boldsymbol{\alpha}(a^*) = \mathrm{do}(T' = 1)$. Therefore, with this alternative, still exact abstraction, it holds that $a'^* \neq \boldsymbol{\alpha}(a^*)$. This difference is due to the symmetries in the base and abstracted models that allow for the preservation of distributions although the domains are swapped.

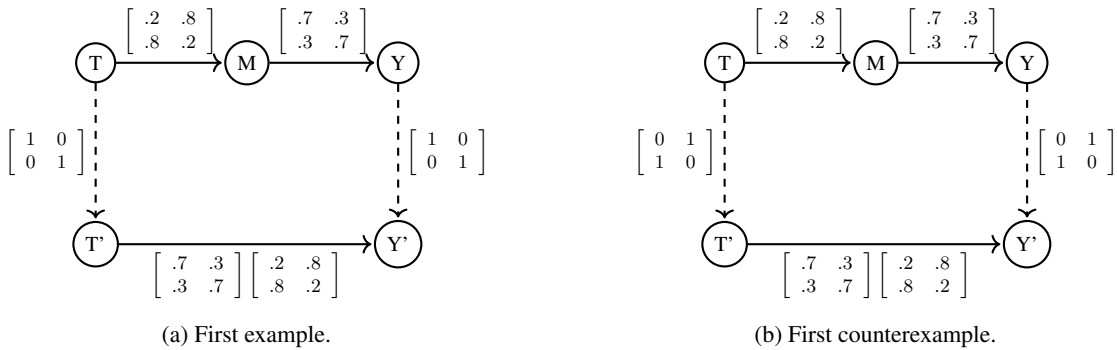

(a) First example.              (b) First counterexample.

Figure 4: Models for the first counterexample.

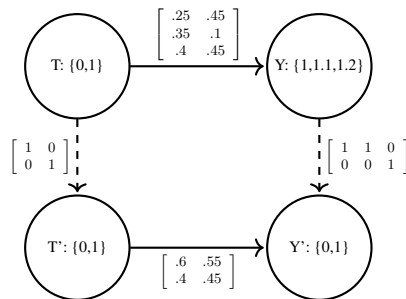

Figure 5: Model for the second counterexample.

*Second counterexample.* In the second counterexample, let $\mathcal{B}$ and $\mathcal{B}'$ be CMABs with the same underlying DAG in Fig. 1(right). The base model $\mathcal{M}$ is defined on a binary variable $T$ and a ternary variable $Y$; we assume the following structural functions $f_T = \begin{bmatrix} .8 \\ .2 \end{bmatrix}$ and $f_Y = \begin{bmatrix} .25 & .45 \\ .35 & .1 \\ .4 & .45 \end{bmatrix}$. We also assign to the domain of $Y$ the following values $\mathbb{D}[Y] = \{1, 1.1, 1.2\}$. In the abstracted model $\mathcal{M}'$ we only use binary variables, and we define the structural functions as $f_{T'} = f_T$ and $f_{Y'} = \begin{bmatrix} .6 & .55 \\ .4 & .45 \end{bmatrix}$. We assign to the domain of $Y'$ the standard values $\mathbb{D}[Y'] = \{0, 1\}$. Finally, let $\boldsymbol{\alpha}$ be defined on $V, m$ as in Example 3.2, and let $\alpha_{T'}$ be the identity while $\alpha_{Y'} = \begin{bmatrix} 1 & 1 & 0 \\ 0 & 0 & 1 \end{bmatrix}$. See Fig. 5 for an illustration.

The outcome node of the base CMAB and the outcome node of the abstracted CMAB are defined over different domains. Notice that the maps $\alpha_{T'}, \alpha_{Y'}$ are order-preserving, preventing us from exploiting symmetries as in the previous counterexample.

Assuming $\mathcal{I} = \{\mathrm{do}(T = 0), \mathrm{do}(T = 1)\}$, the IC error is $e(\boldsymbol{\alpha}) = 0$. The true expected rewards in the base CMAB are $\mathbb{E}_{Y \mid \mathrm{do}(T=0)}[Y] = 1.115$ and $\mathbb{E}_{Y \mid \mathrm{do}(T=1)}[Y] = 1.1$, leading to the optimal action $a^*$ being $\mathrm{do}(T = 0)$.

If we were to use the abstraction $\boldsymbol{\alpha}$ to translate the optimal action $a^*$ we would map it to $\boldsymbol{\alpha}(a^*) = \mathrm{do}(T' = 0)$. However, if we were to compute the true expected rewards in the abstracted CMAB we would get $\mathbb{E}_{Y' \mid \mathrm{do}(T'=0)}[Y'] = 0.4$ and $\mathbb{E}_{Y' \mid \mathrm{do}(T'=0)}[Y'] = 0.45$, meaning that the optimal action $a'^*$ is $\mathrm{do}(T' = 0)$.

Thus, $a'^* \neq \boldsymbol{\alpha}(a^*)$. This difference is due to the different values in the domains of the outcome which, once accounted in the expected rewards, lead to different results. ∎

## C.3  PROOF OF LEMMA 5.4

**Lemma 5.4** (Sufficient condition for preservation of maximum). *Given a CAMAB, the maximum is preserved by the abstraction $\boldsymbol{\alpha}$ is $e(\boldsymbol{\alpha}) + s(\boldsymbol{\alpha}) \leq \frac{1}{2} \min_{a \in \mathcal{A}: \Delta(a) > 0} \Delta(a)$.*

*Proof.* From Prop. 4.1 we know that:

$$|\mu_{\text{do}(\mathbf{x})} - \mu'_{\boldsymbol{\alpha}(\text{do}(\mathbf{x}))}| \le e(\boldsymbol{\alpha}) + s(\boldsymbol{\alpha}).$$

This implies that in the abstraction any mean $\mu_a$ may change as much as $e(\boldsymbol{\alpha}) + s(\boldsymbol{\alpha})$.

Let $\mu^*$ be the mean of the optimal action and $\mu_{a_o}$ the mean of the second-best action, that is $a_o = \text{argmin}_{a \in \mathcal{A}:\Delta(a)>0} \Delta(a)$. Then, for the maximum to be preserved, it must hold:

$$(\mu^* - (e(\boldsymbol{\alpha}) + s(\boldsymbol{\alpha}))) - (\mu_{a_o} + (e(\boldsymbol{\alpha}) + s(\boldsymbol{\alpha}))) \quad > 0 \tag{13}$$

$$(\mu^* - \mu_{a_o}) - 2(e(\boldsymbol{\alpha}) + s(\boldsymbol{\alpha})) \quad > 0 \tag{14}$$

$$\frac{1}{2}\Delta(a_o) > e(\boldsymbol{\alpha}) + s(\boldsymbol{\alpha}) \tag{15}$$

Hence, $e(\boldsymbol{\alpha}) + s(\boldsymbol{\alpha}) \le \frac{1}{2}\min_{a \in \mathcal{A}:\Delta(a)>0} \Delta(a)$. ∎

## C.4 LEMMA C.1

**Lemma C.1** (Algebraic sufficient condition for preservation of maximum). *Given a CAMAB with a zero IC error abstraction, the maximum is preserved if there is no $\mathbf{b} \in \mathcal{A}$ such that $\mathbf{y}'\mathbf{A}_{Y'}(\mathbf{a}^* - \mathbf{b})\mathbf{A}_{X'}^+ < 0$, where $A^+$ is the Moore-Penrose pseudo-inverse of $A$.*

*Proof.* Given the optimal action $a^*$, then, maximum-preservation means that for any action $b \in \mathcal{A}$, it holds that $\mu^* > \mu_b$ and $\mu'_{\boldsymbol{\alpha}(a^*)} > \mu_{\boldsymbol{\alpha}(b)}$.

Let us setup the matrix notation: $\mathbf{y} \in 1\times|\mathbb{D}[Y]|, \mathbf{y}' \in 1\times|\mathbb{D}[Y']|$ are the vectors of reward values; $\mathbf{a}^*, \mathbf{b} \in |\mathbb{D}[Y]|\times1, \mathbf{a}', \mathbf{b}' \in |\mathbb{D}[Y']| \times 1$ are the interventional distributions of rewards for actions $a^*, b$ and their corresponding abstracted actions $a', b'$; $\mathbf{A}_{X'} \in |\mathbb{D}[X']| \times |\mathbb{D}[X]|, \mathbf{A}_{Y'} \in |\mathbb{D}[Y']| \times |\mathbb{D}[Y]|$ encode the abstraction matrices $\alpha_{X'}, \alpha_{Y'}$ respectively.

We can now re-express the two condition for optimum preservation above in matrix form:

$$\begin{cases} \mathbf{y}\mathbf{a}^* > \mathbf{y}\mathbf{b} \\ \mathbf{y}'\mathbf{a}' > \mathbf{y}'\mathbf{b}' \end{cases}$$

This means we need the condition:

$$\mathbf{y}'\mathbf{a}' > \mathbf{y}'\mathbf{b}'$$
$$\mathbf{y}'\mathbf{a}' - \mathbf{y}'\mathbf{b}' > 0$$
$$\mathbf{y}'(\mathbf{a}' - \mathbf{b}') > 0$$

to hold for all $b' \in \mathcal{A}'$, or, equivalently, that there is no $b' \in \mathcal{A}'$ such that:

$$\mathbf{y}'(\mathbf{a}' - \mathbf{b}') \le 0$$

We can now redefine the abstracted interventional distributions $\mathbf{a}'$ and $\mathbf{b}'$ as functions of the base interventional distributions:

$$\mathbf{a}' = \mathbf{A}_{Y'}\mathbf{a}^*\mathbf{A}_{X'}^+$$
$$\mathbf{b}' = \mathbf{A}_{Y'}\mathbf{b}\mathbf{A}_{X'}^+,$$

where $A^+$ is the Moore-Penrose pseudo-inverse of $A$.

Substituting, we get the condition

$$\mathbf{y}'(\mathbf{A}_{Y'}\mathbf{a}^*\mathbf{A}_{X'}^+ - \mathbf{A}_{Y'}\mathbf{b}\mathbf{A}_{X'}^+) \le 0$$
$$\mathbf{y}'\mathbf{A}_{Y'}(\mathbf{a}^* - \mathbf{b})\mathbf{A}_{X'}^+ \le 0$$

which we want to hold for no $b \in \mathcal{A}$. ∎

## C.5   PROOF OF PROPOSITION 5.5

**Proposition 5.5** (Asymptotic regret of `TOpt`). *For $T \to \infty$, using `TOpt`, $\bar{R}'_{\text{TOpt}}(T) \to 0$ iff $\boldsymbol{\alpha}(a^*) = a'^*$.*

*Proof.* By running a CMAB algorithm on the base CMAB $\mathcal{B}$, we have that for $T \to \infty$, the algorithm will converge to the optimal action $a^*$, therefore:

$$\lim_{T \to \infty} \bar{R}_{\text{ALG}}(T) = \mathbb{E}_{\text{ALG}} \left[ \Delta(a^{(T)}) \right]$$
$$= \mathbb{E}_{\text{ALG}} [\Delta(a^*)] = 0$$

At the same time $T$, in the abstracted CMAB we get

$$\bar{R}'_{\text{TOpt}}(T) = \mathbb{E}_{\text{ALG}} \left[ \Delta(a^{(T)}) \right]$$
$$= \Delta(\boldsymbol{\alpha}(a^*))$$

where we lose the expected value because `TOpt` defines a deterministic policy over $\boldsymbol{\alpha}(a^*)$.

Now, because of the assumed preservation of the maximum:

$$\bar{R}'_{\text{TOpt}}(T) = \Delta(\boldsymbol{\alpha}(a^*))$$
$$= \Delta(a'^*) = 0$$

proving that, as the simple regret of the base CMAB converges to zero, so does the simple regret of the abstracted CMAB. ∎

## C.6   PROOF OF LEMMA 5.7

**Lemma 5.7** (Unbiasedness of `IMIT`) *Under the coverage assumption that $\pi^{(t)}$ has non-zero probability for every action $a^{(t)}$, the estimates $\hat{\mu}_{\text{do}(\mathbf{x})}$ of `IMIT` are unbiased.*

*Proof.* In `IMIT`, rewards samples are unbiased since the abstracted agent takes action $\boldsymbol{\alpha}(a^{(t)})$ and receives the reward

$$g'^{(t)} \sim G_{\boldsymbol{\alpha}(a^{(t)})}$$

sampled from the actual abstracted environment of $\mathcal{M}'$.

Now, the requirement that $\pi^{(t)}$ has non-zero probability for every action $a^{(t)}$, together with the surjectivity in assumption (AB1c), implies that:

$$\forall a'_i \in \mathcal{A}' \quad \exists a_i \in \mathcal{A} \quad \text{s.t. } \boldsymbol{\alpha}(a_i) = a'_i,$$

and, therefore:

$$\pi'^{(t)}(a'_i) = \pi^{(t)}(\boldsymbol{\alpha}(a_i)) \neq 0$$

for all actions $a'_i \in \mathcal{A}'$. This means that every action $a' \in \mathcal{I}'$ will be taken with some probability. Thus, in the long run, the expected values of the interventional distributions $\mathbb{P}(Y' | \text{do}(\mathbf{x}_i))$ can be estimated in an unbiased way. ∎

## C.7   PROOF OF PROPOSITION 5.8

**Proposition 5.8** (Confidence of `IMIT`). *Given a CAMAB, assume we have run `UCB` for $T$ steps on $\mathcal{M}$. For `IMIT` to reach the same level of confidence in $\hat{\mu}_{a'_i}$ as running `UCB` for $T$ steps on $\mathcal{M}'$, it must hold $N(\mathcal{K}(a'_i) - 1) + \left( \sum_{a_j \in \boldsymbol{\alpha}^{-1}(a'_i)} \frac{1}{\Delta(a_j)^2} - \frac{1}{\Delta(a'_i)^2} \right) \geq 0$ where $N > 0$ is a constant term.*

*Proof.* Assume we run the `UCB` algorithm on the abstracted CMAB $\mathcal{B}'$. The number of times `UCB` will test an action in order to estimate its mean is proportional to:

$$\mathbb{E}_{\text{UCB}} [\mathcal{C}(a'_i)] \propto N + \frac{1}{\Delta(a'_i)^2}, \tag{16}$$

where the first term is a constant $N$ while the second term depends on the squared action gap.

Assume also that we are running `UCB` on the base CMAB $\mathcal{B}'$ as well; then, under `IMIT`, the number of times that an arm will be pulled will be:

$$\mathbb{E}_{\texttt{IMIT}}[\mathcal{C}(a_i')] \propto \sum_{a_j \in \mathcal{A}: \boldsymbol{\alpha}(a_j)=a_i'} \left(N + \frac{1}{\Delta(a_j)^2}\right)$$

To achieve an estimate of the expected return with the same level of confidence as running `UCB` direcly on $\mathcal{B}'$, it is necessary for `IMIT` that the following relation hold:

$$\sum_{a_j \in \mathcal{A}: \boldsymbol{\alpha}(a_j)=a_i'} \left(N + \frac{1}{\Delta(a_j)^2}\right) \geq N + \frac{1}{\Delta(a_i')^2}$$

that is:

$$\sum_{a_j \in \mathcal{A}: \boldsymbol{\alpha}(a_j)=a_i'} \left(N + \frac{1}{\Delta(a_j)^2}\right) - N + \frac{1}{\Delta(a_i')^2} \;\geq\; 0$$

$$N \left(\sum_{a_j \in \mathcal{A}: \boldsymbol{\alpha}(a_j)=a_i'} 1 - 1\right) + \sum_{a_j \in \mathcal{A}: \boldsymbol{\alpha}(a_j)=a_i'} \frac{1}{\Delta(a_j)^2} - \frac{1}{\Delta(a_i')^2} \;\geq\; 0$$

$$N(\mathcal{K}(a_i') - 1) + \left(\sum_{a_j \in \mathcal{A}: \boldsymbol{\alpha}(a_j)=a_i'} \frac{1}{\Delta(a_j)^2} - \frac{1}{\Delta(a_i')^2}\right) \;\geq\; 0$$

where, $\mathcal{K}(a_i')$ is the size of cluster mapping to $a_i'$, that is, $\sum_{a_j \in \mathcal{A}: \alpha(a_j)=a_i'} 1$. ∎

## C.8   PROOF OF LEMMA 5.9

**Lemma 5.9** (Cumulative regret for `IMIT`). *Given a CAMAB, sssume we have run `ALG` for $T$ steps on $\mathcal{M}$. The difference in cumulative regret between running `ALG` or `IMIT` on $\mathcal{M}'$ is* $\mathbb{E}_{\texttt{ALG}}\left[\sum_{t=0}^{T} \sum_{a_i' \in \mathcal{A}'} \boldsymbol{\alpha}(\mathbb{P}(a'^{(t)} = a_i'))\mu_{a_i'}\right] - \mathbb{E}_{\texttt{ALG}}\left[\sum_{t=0}^{T} \sum_{a_i' \in \mathcal{A}'} \mathbb{P}(a'^{(t)} = a_i')\mu_{a_i'}\right].$

*Proof.* By definition the cumulative regret when learning in the abstracted CMAB using algorithm `ALG` is:

$$R'(T) = \mathbb{E}_{\texttt{ALG}}\left[\sum_{j=t}^{T} \sum_{a_i' \in \mathcal{A}'} \mathbb{P}(a'^{(j)} = a_i')\Delta(a_i')\right].$$

This quantity may be decomposed as:

$$R'(T) = \mathbb{E}_{\texttt{ALG}}\left[\sum_{j=t}^{T} \sum_{a_i' \in \mathcal{A}'} \mathbb{P}(a'^{(j)} = a_i')(\mathbb{E}_{G_{a'*}}[G] - \mathbb{E}_{G_{a_i'}}[G])\right]$$

$$= (T - t)\,\mathbb{E}_{G_{a'*}}[G] - \mathbb{E}_{\texttt{ALG}}\left[\sum_{j=t}^{T} \sum_{a_i' \in \mathcal{A}'} \mathbb{P}(a'^{(j)} = a_i')\mathbb{E}_{G_{a_i'}}[G]\right]$$

For convenience, and with no loss of generality, let's take $t = 0$ and let's redefine the maximal reward that can be accumulated in the abstracted CMAB in $T$ timesteps as $M_T' = (T - 0)\,\mathbb{E}_{G_{a'*}}[G]$. We then have:

$$R'(T) = M_T' - \mathbb{E}_{\texttt{ALG}}\left[\sum_{j=0}^{T} \sum_{a_i' \in \mathcal{A}'} \mathbb{P}(a'^{(j)} = a_i')\mathbb{E}_{G_{a_i'}}[G]\right].$$

In the case of learning via the `IMIT` algorithm `IMIT`, the cumulative regret is:

$$R'_{\text{imit}}(T) = \mathbb{E}_{\texttt{IMIT}}\left[\sum_{j=t}^{T} \sum_{a_i' \in \mathcal{A}'} \mathbb{P}(a'^{(j)} = a_i')\Delta(a_i')\right].$$

Through an analogous decomposition, we can get:

$$R'_{\text{imit}}(T) = M'_T - \mathbb{E}_{\text{IMIT}} \left[ \sum_{j=0}^{T} \sum_{a'_i \in \mathcal{A}'} \mathbb{P}(a'^{(j)} = a'_i) \mathbb{E}_{G_{a'_i}}[G] \right].$$

At each timestep of IMIT, the probability $\mathbb{P}(a'^{(j)} = a'_i)$ of selecting the action $a'_i$ is the same as the probability of algorithm ALG on the base CMAB of selecting an action $a_i$ such that $\boldsymbol{\alpha}(a_i) = a'_i$. Thus, the cumulative regret can be re-expressed as:

$$R'_{\text{imit}}(T) = M'_T - \mathbb{E}_{\text{ALG}} \left[ \sum_{j=0}^{T} \sum_{a_i \in \mathcal{A}} \mathbb{P}(a^{(j)} = a_i) \mathbb{E}_{G_{\boldsymbol{\alpha}(a_i)}}[G] \right].$$

Grouping all actions $a_k$ such that $\boldsymbol{\alpha}(a_k) = a'_i$ we get:

$$R'_{\text{imit}}(T) = M'_T - \mathbb{E}_{\text{ALG}} \left[ \sum_{j=0}^{T} \sum_{a'_i \in \mathcal{A}'} \sum_{a_k | \boldsymbol{\alpha}(a_k) = a'_i} \mathbb{P}(a^{(j)} = a_i) \mathbb{E}_{G_{a'_i}}[G] \right].$$

This, in turn, is just the definition of the pushforward of the distribution $\mathbb{P}(a^{(j)} = a_i)$ via $\boldsymbol{\alpha}$:

$$R'_{\text{imit}}(T) = M'_T - \mathbb{E}_{\text{ALG}} \left[ \sum_{j=0}^{T} \sum_{a'_i \in \mathcal{A}'} \alpha_{X'_i \#}(\mathbb{P})(a'^{(j)} = a'_i) \mathbb{E}_{G_{a'_i}}[G] \right].$$

Now taking the difference $R' - R'_{\text{imit}}$ we obtain:

$$R'(T) - R'_{\text{imit}}(T) = M'_T - \mathbb{E}_{\text{ALG}} \left[ \sum_{j=0}^{T} \sum_{a'_i \in \mathcal{A}'} \mathbb{P}(a'^{(j)} = a'_i) \mathbb{E}_{G_{a'_i}}[G] \right] - M'_T + \mathbb{E}_{\text{ALG}} \left[ \sum_{j=0}^{T} \sum_{a'_i \in \mathcal{A}'} \alpha_{X'_i \#}(\mathbb{P})(a'^{(j)} = a'_i) \mathbb{E}_{G_{a'_i}}[G] \right]$$

$$= \mathbb{E}_{\text{ALG}} \left[ \sum_{j=0}^{T} \sum_{a'_i \in \mathcal{A}'} \alpha_{X'_i \#}(\mathbb{P})(a'^{(j)} = a'_i) \mathbb{E}_{G_{a'_i}}[G] \right] - \mathbb{E}_{\text{ALG}} \left[ \sum_{j=0}^{T} \sum_{a'_i \in \mathcal{A}'} \mathbb{P}(a'^{(j)} = a'_i) \mathbb{E}_{G_{a'_i}}[G] \right],$$

which highlights the role of the abstraction in the difference between the regrets. ∎

### C.9 PROOF OF PROPOSITION 5.10

**Proposition 5.10** (Regret lower bound of IMIT). *Given a CAMAB, assume we have run UCB for $T$ steps on $\mathcal{M}$. For IMIT to have a lower regret bound than running UCB for $T$ steps on $\mathcal{M}'$, it must hold $3 \sum_{a'_i \in \mathcal{A}'} \Delta(a'_i) [1 - \mathcal{K}(a'_i)] +$ $16 \log T \sum_{a'_i \in \mathcal{A}'} \left[ \frac{1}{\Delta(a'_i)} - \Delta(a'_i) \frac{1}{\sum_{\boldsymbol{\alpha}^{-1}(a'_i)} \Delta^2(a_j)} \right] \geq 0.$*

*Proof.* Assume we run UCB on the abstracted CMAB $\mathcal{B}'$. The upper bound on the regret over $T$ timesteps is:

$$R'_{\text{UCB}}(T) \leq \sum_{a'_i \in \mathcal{A}'} \Delta(a'_i) \left[ 3 + \frac{16 \log T}{\Delta^2(a'_i)} \right],$$

Assume also that we are running UCB on the base CMAB $\mathcal{B}'$ as well; then, under IMIT, the regret will be lower bounded by:

$$R'_{\text{IMIT}}(T) \leq \sum_{a'_i \in \mathcal{I}'} \Delta(a'_i) \left[ \sum_{a_j \in \mathcal{A}: \alpha(a_j) = a'_i} 3 + \frac{16 \log T}{\Delta^2(a_j)} \right].$$

For IMIT to achieve a regret lower bound lower than UCB we need:

$$R'_{\text{UCB}}(T) - R'_{\text{IMIT}}(T) \geq \quad 0$$

$$\sum_{a'_i \in \mathcal{A}'} \Delta(a'_i) \left[ 3 + \frac{16 \log T}{\Delta^2(a'_i)} \right] - \sum_{a'_i \in \mathcal{A}'} \Delta(a'_i) \left[ \sum_{a_j \in \mathcal{A}: \alpha(a_j) = a'_i} 3 + \frac{16 \log T}{\Delta^2(a_j)} \right] \geq \quad 0$$

$$3 \sum_{a'_i \in \mathcal{A}'} \Delta(a'_i) + \sum_{a'_i \in \mathcal{A}'} \Delta(a'_i) \frac{16 \log T}{\Delta^2(a'_i)} - \sum_{a'_i \in \mathcal{A}'} \Delta(a'_i) \sum_{a_j \in \mathcal{A}: \alpha(a_j) = a'_i} 3 + \sum_{a'_i \in \mathcal{A}'} \Delta(a'_i) \sum_{a_j \in \mathcal{A}: \alpha(a_j) = a'_i} \frac{16 \log T}{\Delta^2(a_j)} \geq \quad 0$$

Let us define a variable $\mathcal{K}(a'_i)$ that gives us the size of cluster mapping to $a'_i$, that is, $\sum_{a_j \in \mathcal{A}: \alpha(a_j) = a'_i} 1$. Then:

$$3 \sum_{a'_i \in \mathcal{A}'} \Delta(a'_i) + \sum_{a'_i \in \mathcal{A}'} \Delta(a'_i) \frac{16 \log T}{\Delta^2(a'_i)} - \sum_{a'_i \in \mathcal{A}'} \Delta(a'_i) \mathcal{K}(a'_i) 3 + \sum_{a'_i \in \mathcal{A}'} \Delta(a'_i) \sum_{a_j \in \mathcal{A}: \alpha(a_j) = a'_i} \frac{16 \log T}{\Delta^2(a_j)} \geq \quad 0$$

$$3 \left[ \sum_{a'_i \in \mathcal{A}'} \Delta(a'_i) - \sum_{a'_i \in \mathcal{A}'} \Delta(a'_i) \mathcal{K}(a'_i) \right] + 16 \log T \left[ \sum_{a'_i \in \mathcal{A}'} \Delta(a'_i) \frac{1}{\Delta^2(a'_i)} - \sum_{a'_i \in \mathcal{A}'} \Delta(a'_i) \sum_{a_j \in \mathcal{A}: \alpha(a_j) = a'_i} \frac{1}{\Delta^2(a_j)} \right] \geq \quad 0$$

$$3 \sum_{a'_i \in \mathcal{A}'} \Delta(a'_i) \left[ 1 - \mathcal{K}(a'_i) \right] + 16 \log T \left[ \sum_{a'_i \in \mathcal{A}'} \frac{1}{\Delta(a'_i)} - \sum_{a'_i \in \mathcal{A}'} \Delta(a'_i) \frac{1}{\sum_{a_j \in \mathcal{A}: \alpha(a_j) = a'_i} \Delta^2(a_j)} \right] \geq \quad 0$$

$$3 \sum_{a'_i \in \mathcal{A}'} \Delta(a'_i) \left[ 1 - \mathcal{K}(a'_i) \right] + 16 \log T \sum_{a'_i \in \mathcal{A}'} \left[ \frac{1}{\Delta(a'_i)} - \Delta(a'_i) \frac{1}{\sum_{a_j \in \mathcal{A}: \alpha(a_j) = a'_i} \Delta^2(a_j)} \right] \geq \quad 0 \; \blacksquare$$

## C.10  PROOF OF PROPOSITION 5.11

**Proposition 5.11** (Asymptotic regret for `IMIT`) *For $T \to \infty$, the abstracted CMAB using `IMIT` achieves sub-linear cumulative regret iff $\boldsymbol{\alpha}(a^*) = a'^*$.*

*Proof.* By running on the base CMAB $\mathcal{B}$ a CMAB algorithm such as `UCB`, we have that, for $T \to \infty$, the algorithm will converge to the optimal action $a^*$, such that:

$$\mathbb{P}(a^{(T)} = a^*) \to 1$$

Consequently, in the abstracted CMAB we get

$$\mathbb{P}(a'^{(T)} = \boldsymbol{\alpha}(a^*)) \to 1.$$

Now, because of the assumed preservation of the maximum, also the abstracted CMAB will converge to the optimal action $a'^*$:

$$\mathbb{P}(a'^{(T)} = a'^*) \to 1,$$

thus proving that the cumulative regret will grow sub-linearly. $\blacksquare$

## C.11  PROOF OF PROPOSITION 5.13

**Proposition 5.13** (Bias of $\alpha_{\mathbb{E}}$) *Assuming a linear interpolating function $\alpha_{\mathbb{E}}$, the difference $|\alpha_{\mathbb{E}}(\mu_{\text{do}(\mathbf{x})}) - \mu'_{\boldsymbol{\alpha}(\text{do}(\mathbf{x}))}|$ is upper bounded by $|\mathbb{E}_{Y \mid \text{do}(\mathbf{x})}[\epsilon_{Y'}(Y)]| + e(\boldsymbol{\alpha})$, where $\epsilon_{Y'}(Y)$ is the interpolation error introduced by $\alpha_{\mathbb{E}}$.*

*Proof.* We can derive the bound on the difference between the upper path $\alpha_{\mathbb{E}}(\mathbb{E}_{Y \mid \text{do}(\mathbf{x})}[Y])$ and the lower path $\mathbb{E}_{Y' \mid \alpha_{\mathbf{X}'}(\text{do}(\mathbf{x}))}[Y']$ as in Fig. 2c by bounding the following transformations:

$$\alpha_{\mathbb{E}}(\mathbb{E}_{Y \mid \text{do}(\mathbf{x})}[Y]) \to \tag{17}$$

$$\mathbb{E}_{Y \mid \text{do}(\mathbf{x})}[\alpha_{\mathbb{E}}(Y)] \to \tag{18}$$

$$\mathbb{E}_{Y \mid \text{do}(\mathbf{x})}[\alpha_{Y'}(Y)] \to \tag{19}$$

$$\mathbb{E}_{Y' \mid \alpha_{X'}(\text{do}(\mathbf{x}))}[Y']. \tag{20}$$

To evaluate these transformations we will rely on the stated assumption:

(AS1) *linear interpolation function:* we will assume that $\alpha_{\mathbb{E}}$ is learned from a family of linear functions $\mathcal{F}$;

(AS2) *identity of domains:* we will expect the outcome domains in the base and abstracted model to be the same $\mathbb{D}[Y] = \mathbb{D}[Y']$; notice that, if that is not the case, we can always redefine the domains of the outcomes to be the union $\mathbb{D}[Y] \cup \mathbb{D}[Y']$ of the domains. This assumptions guarantees that expected values may be equally computed on the domain of $Y$ or $Y'$.

(i) Let us consider the passage from Eq. 17 to Eq. 18 and let us bound the difference $|\alpha_{\mathbb{E}}(\mathbb{E}_{Y|\operatorname{do}(\mathbf{x})}[Y]) - \mathbb{E}_{Y|\operatorname{do}(\mathbf{x})}[\alpha_{\mathbb{E}}(Y)]|$. We know that, in general, for a function $f$ with second-derivative bounded by $M$, we have:

$$|\mathbb{E}_p[f(X)] - f(\mathbb{E}_p[X])| \leq M\mathbb{V}_p[X],$$

where $p$ is the distribution of $X$. Assuming our extension $\alpha_{\mathbb{E}}$ has bounded second derivative, we get that:

$$|\mathbb{E}_{Y|\operatorname{do}(\mathbf{x})}[\alpha_{\mathbb{E}}(Y)] - \alpha_{\mathbb{E}}(\mathbb{E}_{Y|\operatorname{do}(\mathbf{x})}[Y])| \leq M\mathbb{V}_Y[Y]$$

More specifically, under the assumption (AS1) of a linear extension, because of the zero second derivative, we get that:

$$|\mathbb{E}_{Y|\operatorname{do}(\mathbf{x})}[\alpha_{\mathbb{E}}(Y)] - \alpha_{\mathbb{E}}(\mathbb{E}_{Y|\operatorname{do}(\mathbf{x})}[Y])| \leq 0.$$

Hence, $\mathbb{E}_{\alpha_{\mathbb{E}}(Y|\operatorname{do}(\mathbf{x}))}[Y] = \mathbb{E}_{Y|\operatorname{do}(\mathbf{x})}[\alpha_{\mathbb{E}}(Y)]$. If assumption (AS1) were not to hold, then the passage from Eq. 17 to Eq. 18 would add a contribution greater than zero to the bound.

(ii) Let us consider the passage from Eq. 18 to Eq. 19 and let us bound the difference $|\mathbb{E}_{Y|\operatorname{do}(\mathbf{x})}[\alpha_{\mathbb{E}}(Y)] - \mathbb{E}_{Y|\operatorname{do}(\mathbf{x})}[\alpha_{Y'}(Y)]|$. Notice that in the expression $\mathbb{E}_{Y|\operatorname{do}(\mathbf{x})}[\alpha_{Y'}(Y)]|$ the expected value is taken on the domain of $Y'$ w.r.t. an interventional distribution over $Y$; this expression makes sense thanks to assumption (AS3).

Now, we know that:

$$\alpha_{Y'}(Y) = \alpha_{\mathbb{E}}(Y) + \epsilon_{Y'}(Y),$$

where $\epsilon_{Y'}(Y)$ is the interpolation error due to the approximation of $\alpha_{Y'}$ as $\alpha_{\mathbb{E}}$. We then have:

$$|\mathbb{E}_{Y|\operatorname{do}(\mathbf{x})}[\alpha_{\mathbb{E}}(Y)] - \mathbb{E}_{Y|\operatorname{do}(\mathbf{x})}[\alpha_{Y'}(Y)]| =$$
$$|\mathbb{E}_{Y|\operatorname{do}(\mathbf{x})}[\alpha_{\mathbb{E}}(Y)] - \mathbb{E}_{Y|\operatorname{do}(\mathbf{x})}[\alpha_{\mathbb{E}}(Y) + \epsilon_{Y'}(Y)]| =$$
$$|\mathbb{E}_{Y|\operatorname{do}(\mathbf{x})}[\alpha_{\mathbb{E}}(Y)] - \mathbb{E}_{Y|\operatorname{do}(\mathbf{x})}[\alpha_{\mathbb{E}}(Y)] - \mathbb{E}_{Y|\operatorname{do}(\mathbf{x})}[\epsilon_{Y'}(Y)]| =$$
$$|\mathbb{E}_{Y|\operatorname{do}(\mathbf{x})}[\epsilon_{Y'}(Y)]|.$$

Thus, the bound depends on the quality of the interpolation $\alpha_{\mathbb{E}}$ under the expectation $\mathbb{E}$. In other words, the bounds depends the interpolation error $\epsilon_{Y'}(Y)$ weighted by the distribution over $Y$. An extension $\alpha_{\mathbb{E}}$ that perfectly interpolates all the points of $\alpha_{Y'}$ would reduce this bound to zero.

(iii) Let us consider the passage from Eq. 19 to Eq. 20 and let us bound the difference $|\mathbb{E}_{Y|\operatorname{do}(\mathbf{x})}[\alpha_{Y'}(Y)] - \mathbb{E}_{Y'|\alpha_{X'}(\operatorname{do}(\mathbf{x}))}[Y']|$.

First, because of the definition of pushforward, it holds that, in general, $\mathbb{E}_{f\#p}[X] = \mathbb{E}_p[f(X)]$, therefore:

$$\mathbb{E}_{Y|\operatorname{do}(\mathbf{x})}[\alpha_{Y'}(Y)] = \mathbb{E}_{\alpha_{Y'\#}(Y|\operatorname{do}(\mathbf{x}))}[Y],$$

which, using our shorthanding, can be written as:

$$\mathbb{E}_{Y|\operatorname{do}(\mathbf{x})}[\alpha_{Y'}(Y)] = \mathbb{E}_{\boldsymbol{\alpha}(Y|\operatorname{do}(\mathbf{x}))}[Y].$$

We then want to evaluate the new difference:

$$|\mathbb{E}_{\boldsymbol{\alpha}(Y|\operatorname{do}(\mathbf{x}))}[Y] - \mathbb{E}_{Y'|\alpha_{X'}(\operatorname{do}(\mathbf{x}))}[Y']|.$$

Because of the identity between the outcome domains of assumption (AS3), we can re-express the above difference as:

$$|\mathbb{E}_{\boldsymbol{\alpha}(Y|\operatorname{do}(\mathbf{x}))}[Y] - \mathbb{E}_{Y'|\alpha_{X'}(\operatorname{do}(\mathbf{x}))}[Y]|.$$

Since the Wasserstein distance provides us with a bound on the distance between the expected distributions we have that:

$$|\mathbb{E}_{\boldsymbol{\alpha}(Y\mid\mathrm{do}(\mathbf{x}))}[Y] - \mathbb{E}_{Y'\mid\alpha_{X'}(\mathrm{do}(\mathbf{x}))}[Y]| \leq \mathrm{D}_{\mathrm{W}_2}(\boldsymbol{\alpha}(\mathbb{P}(Y\mid\mathrm{do}(\mathbf{x}))), \mathbb{P}(Y'\mid\alpha_{X'}(\mathrm{do}(\mathbf{x})))),$$

which, again, is bounded by the worst-case distance computed by the IC error over the intervention set:

$$|\mathbb{E}_{\boldsymbol{\alpha}(Y\mid\mathrm{do}(\mathbf{x}))}[Y] - \mathbb{E}_{Y'\mid\alpha_{X'}(\mathrm{do}(\mathbf{x}))}[Y]| \leq e(\boldsymbol{\alpha}).$$

Thus, overall, we can bound the error of transferring the expected reward as:

$$|\alpha_{\mathbb{E}}(\mathbb{E}_{Y\mid\mathrm{do}(\mathbf{x})}[Y]) - \mathbb{E}_{Y'\mid\alpha_{\mathbf{X'}}(\mathrm{do}(\mathbf{x}))}[Y']| \leq |\mathbb{E}_{Y\mid\mathrm{do}(\mathbf{x})}[\epsilon_{Y'}(Y)]| + e(\boldsymbol{\alpha}),$$

as a function of the approximation error and the IC error. $\blacksquare$

## C.12   PROOF OF LEMMA 5.14

**Lemma 5.14** (Confidence bound for $\alpha_{\mathbb{E}}$) *Assuming* $\mathbb{D}[Y] = \mathbb{D}[Y'] = [0,1]$ *and assuming we used a linear interpolating function* $\alpha_{\mathbb{E}}$ *to compute* $\hat{\mu}'_{\mathrm{do}(\mathbf{x}'_i)}$ *as in Eq. 11, with probability at least* $1 - \delta$, *it holds that* $|\mu'_{\mathrm{do}(\mathbf{x}'_i)} - \hat{\mu}'_{\mathrm{do}(\mathbf{x}'_i)}| \leq \kappa$, *where* $\kappa = \sqrt{\frac{2\log(2/\delta)}{\mathcal{C}(\mathrm{do}(a'_i))}} + |\mathbb{E}_{Y\mid\mathrm{do}(\mathbf{x})}[\epsilon_{Y'}(Y)]| + e(\boldsymbol{\alpha})$ *and* $\mathcal{C}(\mathrm{do}(\mathbf{x}'_i))$ *counts the number of times action* $\mathrm{do}(\mathbf{x}'_i)$ *was taken.*

*Proof.* Combining Hoeffding's inequality:

$$\mathbb{P}(|\mathbb{E}_{Y'\mid\mathrm{do}(a'_i)}[Y'] - \hat{\mathbb{E}}_{Y'\mid\mathrm{do}(a'_i)}[Y']| \geq \delta) \leq \exp^{-2\mathcal{C}(a'_i)^2\delta^2}$$

with the bias computed in Prop. 5.13, we can immediately derive a confidence bound given by the sum of the upper bound provided by Hoeffding's inequality [Lattimore et al., 2016] and the bias due to the IC and interpolation error:

$$|\mathbb{E}_{Y'\mid\mathrm{do}(a'_i)}[Y'] - \hat{\mathbb{E}}_{Y'\mid\mathrm{do}(a'_i)}[Y']| \leq \sqrt{\frac{2\log(2/\delta)}{\mathcal{C}(\mathrm{do}(a'_i))}} + |\mathbb{E}_{Y\mid\mathrm{do}(\mathbf{x})}[\epsilon_{Y'}(Y)]| + e(\boldsymbol{\alpha}),$$

that is,

$$|\mathbb{E}_{Y'\mid\mathrm{do}(a'_i)}[Y'] - \hat{\mathbb{E}}_{Y'\mid\mathrm{do}(a'_i)}[Y']| \leq \kappa. \blacksquare$$

## C.13   PROOF OF PROPOSITION 5.15

**Proposition 5.15** (Cumulative regret of TExp) *Given a CAMAB, assume we use* TExp *to initialize the expected rewards of the abstracted CMAB, and then we run* UCB *for $T$ steps. Then the cumulative regret we incur is bounded by:*

$$R'_{\mathrm{TExp}} \leq 3 \sum_{a'_i \in \mathcal{I}'_+} \Delta(a'_i) + \sum_{a'_i \in \mathcal{I}'_+ : \Delta(()a'_i) > 0} \frac{16 + log(T)}{\Delta(a'_i)}.$$

*Proof.* Since TExp is running UCB after transferring the expected values, its cumulative regret equals UCB's cumulative regret [Lattimore and Szepesvári, 2020] after correcting for the new action set $\mathcal{I}'_+$. Therefore:

$$R'_{\mathrm{TExp}} \leq 3 \sum_{a'_i \in \mathcal{I}'_+} \Delta(a'_i) + \sum_{a'_i \in \mathcal{I}'_+ : \Delta(()a'_i) > 0} \frac{16 + log(T)}{\Delta(a'_i)}. \blacksquare$$

# D   CAMAB ALGORITHMS

In this section we summarize the algorithms we have considered, we provide their pseudo-code and we compare them from a computational point of view.

**Direct training.**   As a baseline for our CAMABs, we considered the possibility of training the abstracted agent directly on the CMAB $\mathcal{B}'$ using a standard MAB algorithm ALG (see Alg. 4). In our experiments we adopted the standard UCB algorithm [Lattimore and Szepesvári, 2020]. Direct training requires an agent to select actions, collect rewards, and update its estimates.

---

**Algorithm 4** Direct Learning

---

1: **Input:** CMAB $\mathcal{B}$, time horizon $T$
2: **Output:** optimal policy $\pi$
3: Initialize expected rewards $\hat{\mathbb{E}}_{Y|a_i}^{(0)}[Y]$, auxiliary statistics $\hat{\mathcal{S}}^{(0)}$, and policy $\pi^{(0)}$ {Setup the params}
4: **for** $t = 1...T$ **do**
5:      Select $a^{(t)} \sim \pi^{(t-1)}$ {Decision-making}
6:      Receive $g^{(t)} \sim G_{a^{(t)}}$ {Reward-collection}
7:      Compute $\hat{\mathbb{E}}_{Y|a^{(t)}}^{(t)}[Y], \hat{\mathcal{S}}^{(t)} \leftarrow \text{update}\left(\hat{\mathbb{E}}_{Y|a^{(t)}}^{(t-1)}[Y], \hat{\mathcal{S}}^{(t-1)}, a^{(t)}, g^{(t)}\right)$ {Update stats}
8:      Compute $\pi^{(t)} \leftarrow \text{ALG}\left(\hat{\mathbb{E}}_{Y|a^{(t)}}^{(t)}[Y], \hat{\mathcal{S}}^{(t)}\right)$ {Update policy}
9: **end for**
10: **Return:** $\pi^{(T)}$

---

| | Decision-making | Reward-collection | Update | Action-translation | Reward-translation | Exp value-translation |
|---|---|---|---|---|---|---|
| *Direct* | ✓ | ✓ | ✓ | | | |
| TOpt | | | | ✓ | | |
| IMIT | | ✓ | ✓ | ✓ | | |
| TExp | ✓ | ✓ | ✓ | | | ✓ |

Figure 6: Computational steps

**TOpt.** A naive approach to solve a CAMAB would be to solve the base CMAB $\mathcal{B}$, learn the optimal action $a_o$, and then simply set the policy of the abstracted CMAB $\mathcal{B}'$ to select deterministically the abstracted action $\boldsymbol{\alpha}(a_o)$. The TOpt algorithm does not require any decision-making, reward-collection or update; it just requires the translation of a single action.

**IMIT.** Given the set of trajectories $\mathcal{D} = \{(a^{(t)}, g^{(t)})\}_{t=1}^T$ collected during the training of the base agent, the abstracted agent can be trained by imposing that it follows the corresponding action trajectories under $\boldsymbol{\alpha}$. At each timestep $t$, instead of choosing action $a'^{(t)}$, the agent translates action $a^{(t)}$ using the map $\boldsymbol{\alpha}$ into the image action $\boldsymbol{\alpha}(a^{(t)})$ (see Alg. 2). IMIT removes decision-making from the abstracted agent, but it still requires running the model $\mathcal{M}'$ to compute rewards, as well as updating estimates after every action.

**TExp.** Instead of relying on the trajectories, the abstracted agent could be directly initialized by transferring the expected reward learned by the base agent, discarding sub-optimal action, and then following a direct training approach (see Alg. 3). In the TExp algorithm, the abstracted agent does not have to translate all the trajectories in $\mathcal{D}$, but simply transfer the derived statistics.

**Comparison of the algorithms** The computational steps of algorithms proposed above are summarized in Tab. 6. A couple of observations are in order. The TOpt algorithm is computationally extremely cheap but, as discussed in the main text, it may not return an optimal policy. The IMIT algorithm may be computationally and economically more efficient than direct training, as it replaces the step of decision-making with a simple function application. In addition, when coupled with a learning algorithm ALG where updating statistics $\mathcal{S}^{(t)}$ is not timestep dependent, the IMIT algorithm can process all the trajectories in $\mathcal{D}$ at once in parallel. TExp allows to transfer information more efficiently by translating a single statistics instead of re-running every action; the algorithm follow the steps of a standard MAB algorithm with the advantage of an effective initialization which can allow it to converge to the optimal action faster.

## D.1 TRANSFER OF REWARDS

An intermediate approach between IMIT and TExp is offered by an algorithm transferring rewards. Assuming rewards have been collected and stored, they could be individually transferred through the map $\alpha_{Y'}$, as illustrated by the green arrows in Fig. 2b. This algorithm would transfer individual quantities like IMIT while focusing directly on rewards like TExp.

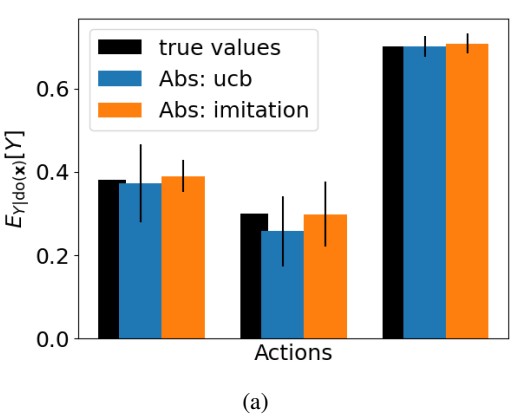
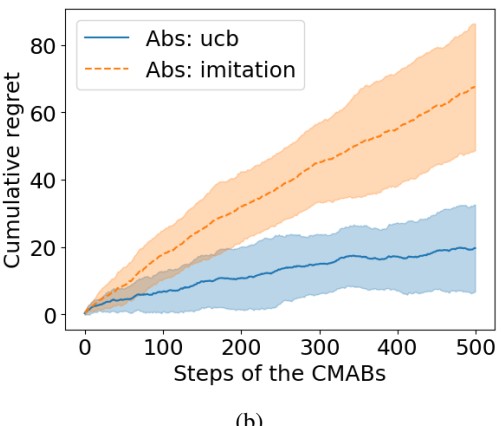

(a)                 (b)

Figure 7: Expected reward and regret using `IMIT`. Shaded areas in (b) represent the standard deviation of the regret.

If given a zero IC error abstraction, reward samples from $\mathbb{P}(Y \mid \mathrm{do}(\mathbf{x}_i))$ can be exactly transformed via $\boldsymbol{\alpha}$ into samples of $\boldsymbol{\alpha}(\mathbb{P}(Y \mid \mathrm{do}(\mathbf{x}_i))) = \mathbb{P}(Y' \mid \boldsymbol{\alpha}(\mathrm{do}(\mathbf{x}_i)))$, allowing for an unbiased estimation of the expected values.

Compared to `TExp`, transporting reward enjoys a lower bias since it relies on the true abstraction map $\alpha_{Y'}$ instead of an extension $\alpha_{\mathbb{E}}$ introducing an interpolation error. Confidence bounds for this algorithm, derived as in Lemma 5.14, could then reduce to $\kappa = \sqrt{\frac{2 \log(2/\delta)}{\mathcal{C}(\mathrm{do}(a'_i))}} + |\mathbb{E}_{Y \mid \mathrm{do}(\mathbf{x})}[\epsilon_{Y'}(Y)]| + e(\boldsymbol{\alpha})$, and used to construct a reduced intervention set $\mathcal{I}'_{+}$.

# E   EXPERIMENTAL DETAILS

## E.1   TRANSFER OF OPTIMAL ACTION

*Scenario 1.* In the first experiment we consider the CAMAB defined in Ex. 5.2 where we take $\alpha_{T'}$ and $\alpha_{Y'}$ to be identity matrices, as illustrated in Fig. 4a. This constitutes a CAMAB with an abstraction which is exact $e(\boldsymbol{\alpha}) = 0$ and maximum preserving. We run `UCB` on the base CMAB for a variable number of steps $n_{steps} = \{10, 25, 50, 100, 250, 500\}$. After $n_{steps}$ we transfer the learned optimal action $a_o$ in the base CMAB to the abstracted CMAB as $\boldsymbol{\alpha}(a_o)$ using `TOpt`. We then compute the simple regret at each $n_{steps}$. We repeat the procedure 20 times and report means of the simple regret as solid lines. The resulting simple regrets are plotted in Fig. 3a in blue, with the solid line representing `TOpt`, and the dashed line representing `UCB`.

*Scenario 2.* In the second experiment we consider the CAMAB defined in Ex. 5.2 where we take $\alpha_{T'}$ and $\alpha_{Y'}$ to be anti-diagonal matrices, as illustrated in Fig. 4b. This constitutes a CAMAB with an abstraction which is exact $e(\boldsymbol{\alpha}) = 0$ and non-maximum preserving. We run the same protocol as on *Scenario 1*. The resulting simple regrets are plotted in Fig. 3a in green, with the solid line representing `TOpt`, and the dashed line representing `UCB`.

## E.2   TRANSFER OF ACTIONS

*Scenario 3.* In the third experiment we consider the CAMAB defined in Ex. 5.2 except for the mechanism $f_{Y'}$ in the abstracted CMAB that takes the form $f_{Y'} = \begin{bmatrix} .7 & .3 \\ .3 & .7 \end{bmatrix}$. This new model is illustrated in Fig. 8a. The change in the abstracted CMAB induces an IC error $e(\boldsymbol{\alpha}) \approx 0.229$. We run `UCB` on the base CMAB, `UCB` and `IMIT` on the abstracted CMAB for 500 steps, while tracking the cumulative regret. We repeat the procedure 20 times and report means of the simple regret as solid lines. Fig. 7a shows that the estimated expected rewards when training $\mathcal{B}'$ directly with `UCB` (blue) or using `IMIT` (orange) are unbiased w.r.t. the true values (black), as stated by Lemma 5.7. However, Fig. 7b confirms that the cumulative regret for `IMIT` is significantly higher than `UCB` due to agent taking suboptimal actions from $\boldsymbol{\alpha}(\pi)$, as implied by Prop. 5.11.

*Scenario 4.* In the fourth experiment we consider the same CAMAB used in *Scenario 1* and illustrated in Fig. 4a. We run the same protocol as on *Scenario 3*. Fig. 10a shows the cumulative regret of `UCB` and `IMIT` on the abstracted CMAB, while Fig.

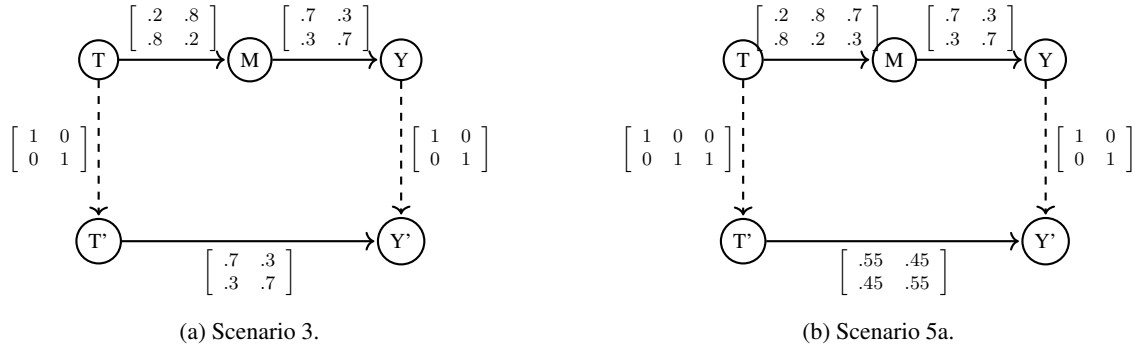

(a) Scenario 3.               (b) Scenario 5a.

Figure 8: Models for experimental simulations.

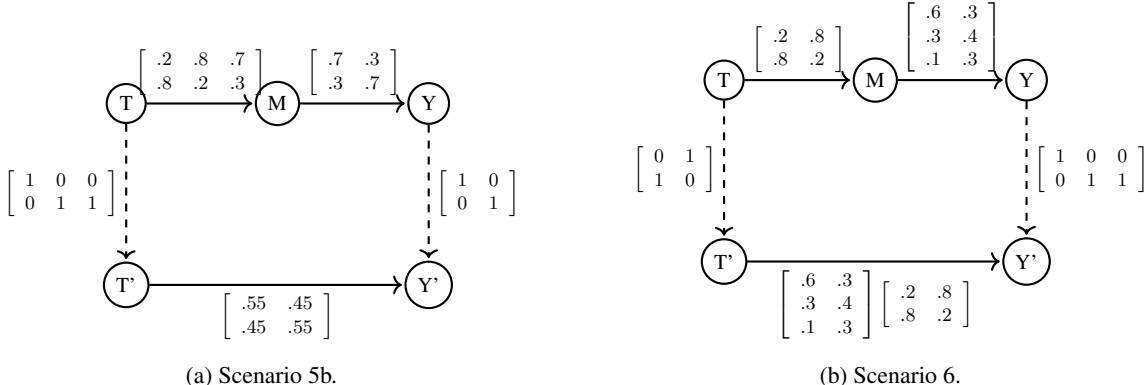

(a) Scenario 5b.               (b) Scenario 6.

Figure 9: Models for experimental simulations.

10b shows the difference between UCB and IMIT in case of zero IC error (*Scenario 4*) and non-zero IC error (*Scenario 3*) in accordance with Lemma 5.9.

*Scenario 5.* In the fifth scenario we instantiate a CAMAB as in Example 3.2. We define the base mechanisms as $f_T = \begin{bmatrix} .7 \\ .2 \\ .1 \end{bmatrix}$, $f_T = \begin{bmatrix} .2 & .8 & .7 \\ .8 & .2 & .3 \end{bmatrix}$, $f_Y = \begin{bmatrix} .7 & .3 \\ .3 & .7 \end{bmatrix}$, and the abstracted mechanisms as $f_{T'} = \begin{bmatrix} .8 \\ .2 \end{bmatrix}$, $f_{Y'} = \begin{bmatrix} .55 & .45 \\ .45 & .55 \end{bmatrix}$.

We consider two possible abstractions: $\boldsymbol{\alpha}_1$ is defined by $V = \{T, Y\}$, $m(T) = T'$, $m(Y) = Y'$, and $\alpha_{T'} = \begin{bmatrix} 1 & 0 & 0 \\ 0 & 1 & 1 \end{bmatrix}$ and $\alpha_{Y'}$ as an identity; $\boldsymbol{\alpha}_2$ is defined in the same way except for $\alpha_{T'} = \begin{bmatrix} 0 & 1 & 0 \\ 1 & 0 & 1 \end{bmatrix}$. The two corresponding CMAB are illustrated in Fig. 8b and 9a. In the base CMAB we consider the following set of action $\mathcal{I} = \{\emptyset, \mathrm{do}(T = 0), \mathrm{do}(T = 1), \mathrm{do}(T = 2)\}$, which is mapped by the abstraction to the set $\mathcal{I}' = \{\emptyset, \mathrm{do}(T' = 0), \mathrm{do}(T' = 1)\}$. The true expected rewards in the base CMAB are $\mathbb{E}_{Y|\emptyset}[Y] = 0.56$, $\mathbb{E}_{Y|\mathrm{do}(T=0)}[Y] = 0.62$, $\mathbb{E}_{Y|\mathrm{do}(T=1)}[Y] = 0.38$, $\mathbb{E}_{Y|\mathrm{do}(T=2)}[Y] = 0.42$, while in the abstracted CMAB we have $\mathbb{E}_{Y'|\emptyset}[Y'] = 0.47$, $\mathbb{E}_{Y'|\mathrm{do}(T'=0)}[Y'] = 0.45$, $\mathbb{E}_{Y'|\mathrm{do}(T'=1)}[Y'] = 0.55$. Thus, in the base CMAB $a^* = \mathrm{do}(T = 0)$ and in the abstracted CMAB $a'^* = \mathrm{do}(T' = 1)$. We run the same protocol as on *Scenario 3*. Fig. 3b shows the difference between the cumulative regret of UCB and IMIT on the abstracted CMAB in the two CAMAB differing only for the abstraction, $\boldsymbol{\alpha}_1$ and $\boldsymbol{\alpha}_2$. Notice that the result is consistent with our theoretical results, as $\boldsymbol{\alpha}_1$ which does not preserve the optimal action achieves much higher cumulative regret than $\boldsymbol{\alpha}_1$ which maps $a^*$ and another high-reward action onto $a'^*$.

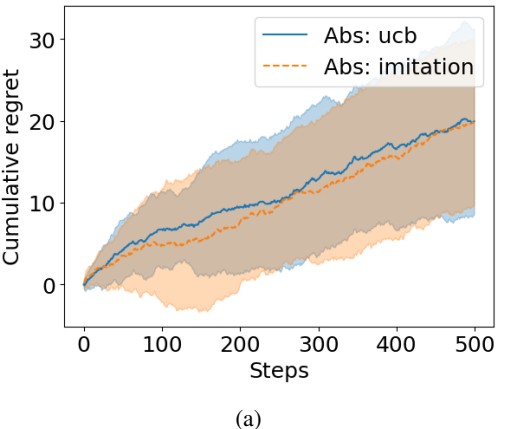
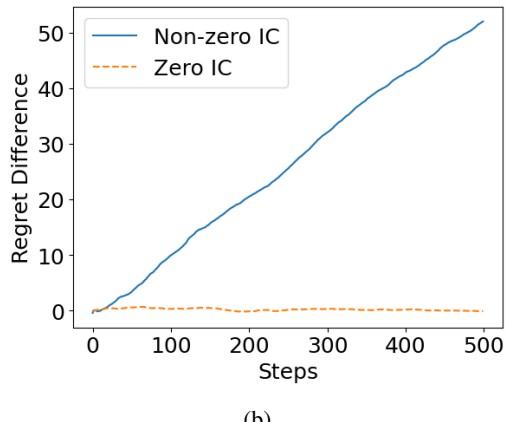

(a)                    (b)

Figure 10: Regret using `IMIT`. Shaded areas in (a) represent the standard deviation of the regret.

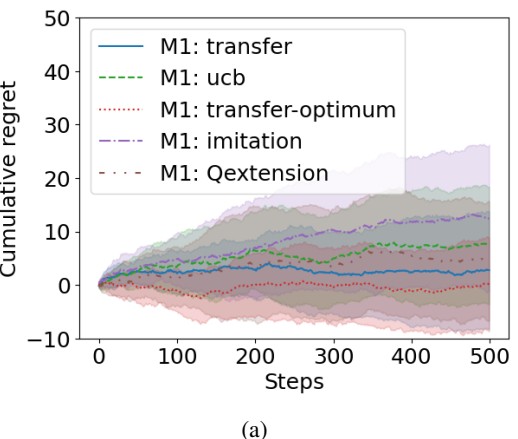
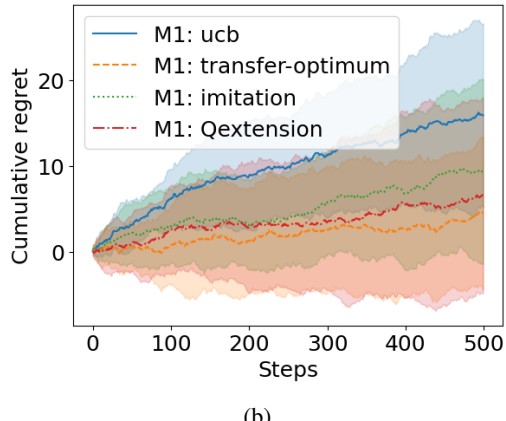

(a)                    (b)

Figure 11: Regret for various CMAB transfer learning schemes. Shaded areas represent the standard deviation of the regret.

### E.3    TRANSFER OF EXPECTED VALUES

*Scenario 6.* In the sixth experiment we consider the CAMAB defined in Ex. 5.2 except for the domains $\mathbb{D}[Y] = \mathbb{D}[Y'] = \{0, 1, 2\}$, the mechanism $f_Y$ in the base CMAB that takes the form $f_Y = \begin{bmatrix} .6 & .3 \\ .3 & .4 \\ .1 & .3 \end{bmatrix}$ and the map $\alpha_{Y'}$ being an identity.

This new model is illustrated in Fig. 9b. The CAMAB has zero IC error. We run `UCB` on the base CMAB, `UCB` and `TExp` on the abstracted CMAB for $500$ steps, while tracking the cumulative regret. We repeat the procedure 20 times and report means of the simple regret as solid lines. Fig. 3c (blue) shows the cumulative regret of `TExp` (solid) and `UCB` (dashed).

*Scenario 7.* In the seventh experiment we consider the CAMAB used in *Scenario 6* but we redefine the domain $\mathbb{D}[Y'] = \{0.4, 0.5, 10\}$. We run the same protocol as on *Scenario 5*. Fig. 3c (green) shows the cumulative regret of `TExp` (solid) and `UCB` (dashed).

### E.4    COMPARISON WITH TRANSFER LEARNING

We compare here our approach to the `B-UCB` algorithm proposed in Zhang and Bareinboim [2017] for transferring information across different CMABs with different but related structures. Zhang and Bareinboim [2017] considers three different tasks; we reproduce the first two tasks, but we exclude the third task, as it would require modelling an abstraction going from an abstracted model to a base model, which we have not worked out yet.

*Task 1.* In the first task the source model $\mathcal{M}$ is defined on three binary variables: the two observable $X, Y$ and the

unobserved $U$. We encode the original mechanisms in the following matrices: $f_U = \begin{bmatrix} .3 \\ .7 \end{bmatrix}$, $f_X(U) = \begin{bmatrix} 1 & 0 \\ 0 & 1 \end{bmatrix}$, and $f_Y(U, X) = \begin{bmatrix} .9 & .5 & .1 & .7 \\ .1 & .5 & .9 & .3 \end{bmatrix}$. Allowed actions are $\mathrm{do}(X = 0)$ and $\mathrm{do}(X = 1)$. The target model $\mathcal{M}'$ is similarly defined on three binary variables: the two observable $X', Y'$ and the unobserved $U'$. The different mechanisms are encoded in the following matrices: $f_{U'} = \begin{bmatrix} .3 \\ .7 \end{bmatrix}$, $f_{X'}(U') = \begin{bmatrix} .5 \\ .5 \end{bmatrix}$, and $f_{Y'}(U', X') = \begin{bmatrix} .9 & .5 & .1 & .7 \\ .1 & .5 & .9 & .3 \end{bmatrix}$. Allowed actions are $\mathrm{do}(X' = 0)$ and $\mathrm{do}(X' = 1)$.

We take the source and target model as the base and abstraction model of an abstraction, respectively. We define a CAMAB setting up an abstraction $\alpha$ with the following natural choices:

- $V = \{U, X, Y\}$;
- $m(U) = U', m(X) = X', m(Y) = Y'$;
- all the matrices $\alpha_{U'}, \alpha_{X'}, \alpha_{Y'}$ as identities.

The map between the intervention sets is automatically defined. Notice that we do not collect observational samples, but only interventional samples for interventions that satisfy assumption (AB1).

Given this setup, we solve the CMABs using standard `UCB`, `B-UCB` [Zhang and Bareinboim, 2017], `TOpt`, `IMIT` and `TExp` for 500 steps. We repeat the training 20 times, and show the cumulative regret in Fig. 11a.

*Task 2*. In the second task the source model $\mathcal{M}$ is defined on four binary variables: the three observable $Z, X, Y$ and the unobserved $U$. We encode the original mechanisms in the following matrices: $f_U = \begin{bmatrix} .2 \\ .8 \end{bmatrix}$, $f_Z = \begin{bmatrix} .1 \\ .9 \end{bmatrix}$, $f_X(U, Z) = \begin{bmatrix} 1 & 0 & 0 & 1 \\ 0 & 1 & 1 & 0 \end{bmatrix}$, and $f_Y(U, X) = \begin{bmatrix} .1 & .9 & .5 & .1 \\ .9 & .1 & .5 & .9 \end{bmatrix}$. Allowed actions are $\mathrm{do}(X = 0)$ and $\mathrm{do}(X = 1)$. The target model $\mathcal{M}'$ is similarly defined on three binary variables: the two observable $X', Y'$ and the unobserved $U'$. The different mechanisms are encoded in the following matrices: $f_{U'} = \begin{bmatrix} .2 \\ .8 \end{bmatrix}$, $f_{X'}(U') = \begin{bmatrix} .5 \\ .5 \end{bmatrix}$, and $f_{Y'}(U', X') = \begin{bmatrix} .1 & .9 & .5 & .1 \\ .9 & .1 & .5 & .9 \end{bmatrix}$. Allowed actions are $\mathrm{do}(X' = 0)$ and $\mathrm{do}(X' = 1)$.

We take the source and target model as the base and abstraction model of an abstraction, respectively. We define a CAMAB setting up an abstraction $\alpha$ with the following natural choices:

- $V = \{U, X, Y\}$;
- $m(U) = U', m(X) = X', m(Y) = Y'$;
- all the matrices $\alpha_{U'}, \alpha_{X'}, \alpha_{Y'}$ as identities.

The map between the intervention sets is automatically defined. Notice that we do not rely on interventional samples, and we focus only on those interventions that comply with our assumption (AB1); therefore, we ignore, possible interventions on $Z$ in the base model $\mathcal{M}$.

Given this setup, we solve the CMABs using standard `UCB`, `B-UCB` [Zhang and Bareinboim, 2017], `TOpt`, `IMIT` and `TExp` for 500 steps. We repeat the training 20 times, and show the cumulative regret in Fig. 11b. Notice the absence of a line for `B-UCB`; this is due to the fact that, relying on the computed bounds, `B-UCB` can immediately identify the optimal action.

With the exception of the cases where `TOpt` achieves a lower regret thanks to the preservation of the maximum via the abstraction, in general, `B-UCB` performs better than `IMIT` and `TExp`. This is due to the fact that `B-UCB` provides tighter bounds than `TExp`. Indeed, the bounds of `B-UCB` are specifically derived for the models under consideration. However, its specificity comes at the cost of the applicability of the algorithm, as discussed in the main paper: while `B-UCB` requires to derive custom bounds between models defined on identical variables,

### E.5  ONLINE ADVERTISEMENT

In the online advertisement simulation we run our algorithm on the CMAB defined in Lu et al. [2020a] to model an advertisement campaign at Adobe. The base CMAB $\mathcal{B} = \langle \mathcal{M}, \mathcal{I} \rangle$ is defined on the endogenous variables $\mathcal{X} =$

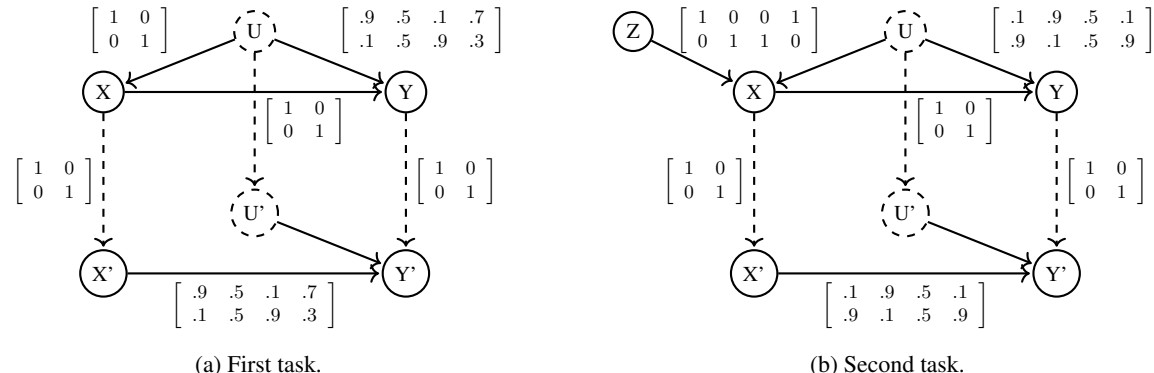

(a) First task.       (b) Second task.

Figure 12: Models for the transfer tasks.

$\{Pr, Pu, SL, BT, ST, CK\}$ with mechanisms as follows:

- $Pr$ is the product advertised, with values in $\mathbb{D}[Pr] = \{$Photoshop, Acrobat IX Pro, Stock$\}$; the mechanism associated with this variable is $f_{Pr} = \begin{bmatrix} .2 \\ .2 \\ .6 \end{bmatrix}$;

- $Pu$ is the purpose of the advertisement, with values in $\mathbb{D}[Pu] = \{$operational, promo, nurture, awareness$\}$; the mechanism associated with this variable is $f_{Pu} = \begin{bmatrix} .05 \\ .6 \\ .3 \\ .05 \end{bmatrix}$;

- $SL$ is the subject length of the sent message, with values in $\mathbb{D}[SL] = \{$less or equal to 7 words, more than 7 words$\}$; the mechanism associated with this variable is $f_{SL}(Pu) = \begin{bmatrix} .3 & .3 & .7 & .7 \\ .7 & .7 & .3 & .3 \end{bmatrix}$;

- $BT$ is the body template of the message, with values in $\mathbb{D}[BT] = \{$template 1, template 2$\}$; the mechanism associated with this variable is $f_{BT}(Pr, Pu) = \begin{bmatrix} .2 & .1 & .5 & .8 & .2 & .1 & .5 & .8 & .4 & .3 & .4 & .5 \\ .8 & .9 & .5 & .2 & .8 & .9 & .5 & .2 & .6 & .7 & .6 & .5 \end{bmatrix}$;

- $ST$ is the sending out time of the message, with values in $\mathbb{D}[ST] = \{$morning, afternoon, evening$\}$; the mechanism associated with this variable is $f_{ST} = \begin{bmatrix} .5 \\ .2 \\ .3 \end{bmatrix}$;

- $CK$ is the clicking of a customer, with values in $\mathbb{D}[CK] = \{0, 1\}$; the mechanism associated with this variable is $f_{CK}(SL, BT, ST) = \begin{bmatrix} 3/9 & 4/9 & 5/9 & 4/9 & 5/9 & 6/9 & 4/9 & 5/9 & 6/9 & 5/9 & 6/9 & 7/9 \\ 6/9 & 5/9 & 4/9 & 5/9 & 4/9 & 3/9 & 5/9 & 4/9 & 3/9 & 4/9 & 3/9 & 2/9 \end{bmatrix}$.

As in Lu et al. [2020a] we take that the only intervenable variables are the choice of the product $(Pr)$, the purpose $(Pu)$ and the sendout time $(ST)$. This CMAB is illustrated in Fig. 13(left).

From this CMAB, we derived a simplified CMAB through the following simplifications: (i) we drop the send-out time variable $(ST)$ since it is not strictly related to the time the mail is read by the customer; (ii) we simplify the number of purposes $(Pu)$, both because some purposes are very unlikely (operational, awareness) and because they have identical effect on subject length $(SL)$; (iii) we reduce the number of products, assuming Photoshop and Stock being merged into a single product. We then instantiate an abstracted CMAB $\mathcal{B}' = \langle \mathcal{M}', \mathcal{I}' \rangle$ defined on the endogenous variables $\mathcal{X} = \{Pr', Pu', SL', BT', ST', CK'\}$, where:

- $Pr'$ has the same meaning of $Pr$ but has domain $\mathbb{D}[Pr'] = \{$Photoshop, Acrobat IX Pro$\}$ and mechanism $f_{Pr'} = \begin{bmatrix} .8 \\ .2 \end{bmatrix}$;

- $Pu'$ has the same meaning of $Pu$ but has domain $\mathbb{D}[Pu'] = \{$promo, nurture$\}$ and mechanism $f_{Pu'} = \begin{bmatrix} .65 \\ .35 \end{bmatrix}$;

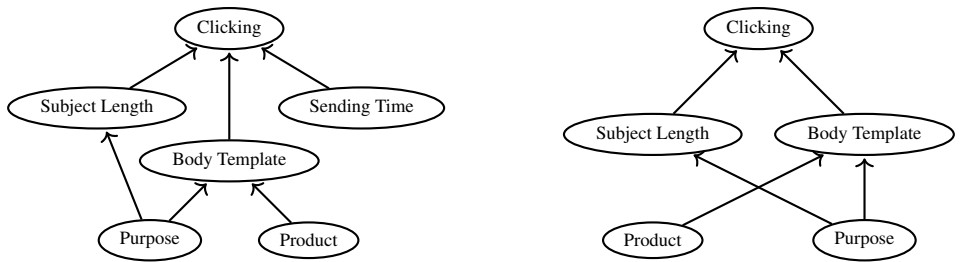

Figure 13: Email campaign CAMAB: base CMAB from Lu et al. [2020a] (left) and abstracted CMAB (right).

- $SL'$ has the same meaning and domain of $SL$, but mechanism $f_{SL'}(Pu') = \begin{bmatrix} .3 & .7 \\ .7 & .3 \end{bmatrix}$;

- $BT'$ has the same meaning and domain of $BT$, but mechanism $f_{BT'}(Pr', Pu') = \begin{bmatrix} .3 & .5 & .15 & .65 \\ .7 & .5 & .85 & .35 \end{bmatrix}$;

- $CK'$ has the same meaning and domain of $CK$, but mechanism $f_{CK'}(SL', BT') = \begin{bmatrix} 5/9 & 4/9 & 4/9 & 3/9 \\ 4/9 & 5/9 & 5/9 & 6/9 \end{bmatrix}$;

The remaining intervenable variables are the choice of the product ($Pr'$), the purpose ($Pu'$). This CMAB is illustrated in Fig. 13(right).

To fully define our CAMAB, we define an abstraction $\alpha$ between the CMABs:

- $V = \{Pr, Pu, SL, BT, CK\}$;
- $m(Pr) = Pr', m(Pu) = Pu', m(SL) = SL', m(BT) = BT', m(CK) = CK'$;
- $\alpha_{Pr'} = \begin{bmatrix} 1 & 0 & 1 \\ 0 & 1 & 0 \end{bmatrix}, \alpha_{Pu'} = \begin{bmatrix} 1 & 1 & 0 & 0 \\ 0 & 0 & 1 & 1 \end{bmatrix}, \alpha_{SL'} = \alpha_{BT'} = \alpha_{CK'} = \begin{bmatrix} 1 & 0 \\ 0 & 1 \end{bmatrix}$.

To satisfy the assumption (AB1) we define the actions in the intervention sets as follows:

- $\mathcal{I} = \{\text{do}(Pu = \text{operational}), \text{do}(Pu = \text{promo}), \text{do}(Pu = \text{nurture}), \text{do}(Pu = \text{awareness}), \text{do}(Pr = \text{Photoshop}), \text{do}(Pr = \text{Acrobat IX Pro})\}$
- $\mathcal{I}' = \{\text{do}(Pu' = \text{promo}), \text{do}(Pu' = \text{nurture}), \text{do}(Pr' = \text{Photoshop}), \text{do}(Pr' = \text{Acrobat IX Pro})\}$

with the mapping between the two action sets naturally given by $\alpha_{Pu'}, \alpha_{Pr'}$.

Now given this CAMAB, we solve the abstracted CMAB running UCB, TOpt, IMIT, and TExp for 1000 episodes. We repeat the procedure 20 times and report means of the simple regret as solid lines. Fig. 3d shows the regret of all our algorithms. The success of TOpt and IMIT can be explained respectively by the preservation of the optimum and the aggregation of the two actions with higher exepcted rewards. The result of TExp is due to a moderate IC error and a low interpolation error; the resulting bounds were not tight enough to exclude actions from $\mathcal{I}'_+$, but sufficient to transfer information to achieve a regret in line with, or inferior to UCB.

## F  FURTHER RELATED WORK

MABs with side information are closely related to the transfer of information between MABs. For example, Zhang and Bareinboim [2017] leverage causal tools to derive upper and lower bounds on the mean reward associated with each action, before running an adapted UCB algorithm which leverages this information. In this sense, the approach proposed by Zhang and Bareinboim [2017] is similar to the algorithm of Sharma et al. [2020] who leverage side information in the form of bounds on the mean reward of each arm to improve regret guarantees.

Similarly, CMABs are closely connected to MABs with correlated arms [Singh et al., 2024, Gupta et al., 2021] in the sense that the SCM underlying a CMAB implicitly describes a correlation structure between actions. Likewise, the CAMAB problem we introduce shares similarities to MABs with side observations [Kocák et al., 2014, Mannor and Shamir, 2011, Caron et al., 2012, Buccapatnam et al., 2014], wherein taking one action provides reward information about other arms.

Regional MABs [Wang et al., 2018a,b] also share a close connection to the transfer setting we propose. In regional MABs, actions are partitioned into clusters indicating that their reward distributions share a common but unknown parameter. Given a causal abstraction exists between two SCMs, interventions in the base model mapped to the same abstracted intervention are implicitly clustered together. However, the meaning of these clusters in terms of their relevance to the expected reward of a given action is far more ambiguous in our setting.

A further review of causal abstraction relating the framework of $\tau$-transformations Rubenstein et al. [2017], Beckers and Halpern [2019] and $\alpha$ abstraction Rischel [2020] is provided in Zennaro [2022]. A related approach to simplifying models based on clustering was proposed by Anand et al. [2023]. Results on measuring abstraction error have been developed for both frameworks in Beckers et al. [2020], Rischel and Weichwald [2021], Zennaro et al. [2023b]. Finally, both abstraction frameworks have been used for learning abstraction between SCMs [Zennaro et al., 2023a, Felekis et al., 2023, Dyer et al., 2023, Xia and Bareinboim, 2024].