# OpenReview forum: "Causally Abstracted Multi-armed Bandits"
_auai.org/UAI/2024/Conference — UAI 2024 oral_

### Official Review · Reviewer_tMcg · 2024-03-20

**Q2-1 Originality-Novelty:** 2
**Q2-2 Correctness-Technical Quality:** 3
**Q2-5 Clarity Of Writing:** 4

**Q10 Ethical Concerns:**

No.

**Q1 Summary And Contributions:**

This paper introduces the causal abstracted MABs problem, aiming to efficiently leverage prior knowledge from a base model to solve the causal bandit on an abstract model. The authors explore various algorithms for this problem and analyze their applicability to different scenarios.

**Q2-3 Extent To Which Claims Are Supported By Evidence:**

4: Excellent: all claims are supported by very convincing evidence (in the form of comprehensive experimental evaluation, rigorous mathematical proofs, detailed (pseudo-)code, precise references, well-motivated and realistic assumptions) and the authors deliver what they promise.

**Q2-4 Reproducibility:**

4: Excellent: key resources (e.g. proofs, code, data) are available and key details (e.g. proof sketches, experimental setup) are comprehensively described for competent researchers to confidently and easily reproduce the main results.

**Q3 Main Strengths:**

1- The paper addresses an important problem that previous approaches have not been able to tackle.
2- The paper considers various approaches and precisely analyzes their applicability. Moreover, experiments conducted on the online advertisement graph showcase the efficiency of the proposed approach.
3- The paper is well-written with a clear flow. Additionally, it thoroughly reviews previous work and distinguishes it from the current study.

**Q4 Main Weakness:**

1- Most of the theoretical results can be seen as either straightforward observations or natural consequences of previous work.

**Q5 Detailed Comments To The Authors:**

1- What is the exact advantage of the IMIT algorithm compared to direct learning? It appears to solely exclude the selection of action in each round.
2- You assume that the action set is finite, resulting in finite domain sizes of variable if we can have atomic intervention. Is it possible to generalize your results to variables with a domain equal to $\mathbb{R}$?

Minor issues:

1- I think Equation 9 is not clear; perhaps there is a better way to illustrate the distance between policies.
2- In the Related work section, the reference in the sentence below is not correct. It should be [Lattimore et al., 2016].
The CMAB problem, in which actions are interventions on an SCM with known causal graph, was formally introduced by Lattimore and Szepesvári [2020], who designed an algo- rithm with sublinear simple regret guarantees.
3- I would be better to transfer the pseudo-codes of the algorithms to the main body of the paper.

**Q9 Complying With Reviewing Instructions:**

Yes

---

> ### Author Rebuttal · Authors · 2024-04-05
>
> We are grateful to the reviewer for their feedback and suggestions. We address the offered comments in the following.
>
> > Most of the theoretical results can be seen as either straightforward observations or natural consequences of previous work.
>
> While we recognize our reliance on existing results in the bandit literature, we would like to stress that we use such results to study and characterize non-trivial and non-intuitive results of a novel bandit formulation for transfer learning. We refer the reviewer also to our answer to reviewer YPfN about the significance of our theoretical contributions.
>
> > What is the exact advantage of the IMIT algorithm compared to direct learning? It appears to solely exclude the selection of action in each round.
>
> From the computational point of view, the advantage of the IMIT algorithm is indeed to exclude the decision-making step (we discuss this in the Appendix too). It is also worth noting that it could be reasonable to expect that delegating decision-making to a more detailed model (the base bandit) with a well-behaving abstraction (zero IC error) would offer a better regret; however, one of our points is to show that this intuitive advantage does not hold in general.
>
> > You assume that the action set is finite, resulting in finite domain sizes of variable if we can have atomic intervention. Is it possible to generalize your results to variables with a domain equal to $\mathbb{R}$?
>
> Our assumption of a finite action set stems mainly from the assumptions of the abstraction framework we have adopted, which is defined over discrete variables. We believe that an extension to continuous actions would be possible, but this would be a direction of future research, as it would require first of all a proper and rigorous extension of the underlying abstraction framework. In addition, extending bandit algorithms, such as UCB, to continuous action spaces necessitates assumptions on the reward distributions associated with each action. Both linearity and Lipschitz continuity of rewards are common assumptions in the bandit literature. We conjecture that similar assumptions on the structural equations of causal models would be necessary to reason about continuous interventions in causal bandit settings.
>
> > I think Equation 9 is not clear; perhaps there is a better way to illustrate the distance between policies.
>
> Equations (8) and (9), in analogy with Equations (6) and (7), are meant to show that a certain identity between two quantities does not imply an identity between two other quantities.
>
> > In the Related work section, the reference in the sentence below is not correct. It should be [Lattimore et al., 2016]. The CMAB problem, in which actions are interventions on an SCM with known causal graph, was formally introduced by Lattimore and Szepesvári [2020], who designed an algorithm with sublinear simple regret guarantees.
>
> We thank the reviewer for pointing the mistake. We corrected the reference.
>
> > It would be better to transfer the pseudo-codes of the algorithms to the main body of the paper.
>
> We agree with the reviewer. As also suggested by reviewer JY7u, we will use the additional space to move the pseudo-code in the main body of the paper.

---

### Official Review · Reviewer_JY7u · 2024-03-21

**Q2-1 Originality-Novelty:** 3
**Q2-2 Correctness-Technical Quality:** 3
**Q2-5 Clarity Of Writing:** 3

**Q1 Summary And Contributions:**

The paper discusses how to transfer information and policies across different Structural Causal Models (SCMs) defined on possibly different variables to represent a Multi-Armed Bandit (MAB) decision making problem. To this end, they formalize the problem by focusing on the scenario where one of the two models is a possibly inexact Causal Abstraction of the other. In this way, they are able to study the error and the regret of computing a solution for the abstract MAB on the lower-level SCM. Overall, the paper highlights the required properties of the abstraction map and that transferring information improves the regret performance compared to directly solving the problem on the abstract model.

**Q2-3 Extent To Which Claims Are Supported By Evidence:**

3: Good: the main claims are supported by convincing evidence (in the form of adequate experimental evaluation, proofs, (pseudo-)code, references, assumptions).

**Q2-4 Reproducibility:**

3: Good: key resources (e.g. proofs, code, data) are available and key details (e.g. proofs, experimental setup) are sufficiently well-described for competent researchers to confidently reproduce the main results.

**Q3 Main Strengths:**

The paper tackles an interesting and novel problem and extensively explore different strategies to solve the Causal Multi-Armed Bandit problem (CMAB) on an abstract SCM. The authors are the first to tackle the problem and propose solid conditions to determine whether transferring information using an abstraction map is beneficial or not for an abstract CMAB. Overall, they prove and show empirically on simulated and real data that there is a clear advantage in transferring information across causally abstracted models.

**Q4 Main Weakness:**

My main concerns lie in the presentation of the proposal, as I further detail in the comments to the authors. Overall, the paper consistently requires to consult the Appendix even to appreciate the core of the contribution, such as the proposed algorithms, which are only reported diagrammaticaly, or the experimental study.

**Q5 Detailed Comments To The Authors:**

- **Figure 3.** There is no legend in Figures 3a and 3c, so the plot is not intelligible without resorting to the main text. Further, while in Appendix E the authors claim to report results averaged over twenty runs with the resulting standard deviation, all subfigures only report a single line. Finally, in Appendix E.1, Scenario 2 is incorrectly referred to as the blue line, while according to the main text, it should be the green one, where `TOpt` does not converge.
- **UCB.** The authors first mention `UCB` in Example 5.6 without any introduction. While it is a standard algorithm for MABs, it could have used at least an introductory sentence and a reference for readers not familiar with the MAB literature.

**Q9 Complying With Reviewing Instructions:**

Yes

---

> ### Author Rebuttal · Authors · 2024-04-05
>
> We thank the reviewer for reading the paper and offering their comments for improvement. In the following we address the comments of the reviewer.
>
> > My main concerns lie in the presentation of the proposal, as I further detail in the comments to the authors. Overall, the paper consistently requires to consult the Appendix even to appreciate the core of the contribution, such as the proposed algorithms, which are only reported diagrammaticaly, or the experimental study.
>
> We appreciate the concern of the reviewer. Whilst the full paper contains all relevant and necessary notions, we understand that reference to the Appendix may hinder ease of reading. With this concern in mind, we will use the additional 2 pages to provide more details and reduce the reliance on the Appendix. In particular, as also requested by reviewer tMcg, we will move the pseudo-code  for each algorithm to the main body.
>
> > Figure 3. There is no legend in Figures 3a and 3c, so the plot is not intelligible without resorting to the main text. Further, while in Appendix E the authors claim to report results averaged over twenty runs with the resulting standard deviation, all subfigures only report a single line. Finally, in Appendix E.1, Scenario 2 is incorrectly referred to as the blue line, while according to the main text, it should be the green one, where TOpt does not converge.
>
> We corrected all these points, and we are grateful to the reviewer for their careful reading and feedback.
>
> > UCB. The authors first mention UCB in Example 5.6 without any introduction. While it is a standard algorithm for MABs, it could have used at least an introductory sentence and a reference for readers not familiar with the MAB literature.
>
> We agree with the reviewer, and we now introduce UCB in the Background section together with a reference.

---

### Official Review · Reviewer_wbEY · 2024-03-22

**Q2-1 Originality-Novelty:** 3
**Q2-2 Correctness-Technical Quality:** 3
**Q2-5 Clarity Of Writing:** 3

**Q1 Summary And Contributions:**

The paper introduces a framework that information between two CMABs can be transferred and exploited for better performance and lower regrets. They propose some theoretical results for guarantee of performance or regret bounds for this framework.

**Q2-3 Extent To Which Claims Are Supported By Evidence:**

3: Good: the main claims are supported by convincing evidence (in the form of adequate experimental evaluation, proofs, (pseudo-)code, references, assumptions).

**Q2-4 Reproducibility:**

3: Good: key resources (e.g. proofs, code, data) are available and key details (e.g. proofs, experimental setup) are sufficiently well-described for competent researchers to confidently reproduce the main results.

**Q3 Main Strengths:**

1. They introduce a framework that information between two CMABs can be transferred and exploited for better performance and lower regrets.
2. They propose some theoretical results for guarantee of performance or regret bounds for this framework.

**Q4 Main Weakness:**

1. Most of their theoretical results are corollaries of similar results in CMAB. The conditions of these theoretical results should be illustrated in example or experiments for justification.
2. Some of definitions are a bit unclear: the procedure and target of CAMAB "problem" is not clearly given. Do the three types of CAMAB settings in section 5 aim for the same problem?

**Q5 Detailed Comments To The Authors:**

If there is a common cause $U$ of treatment and outcome before abstraction, and it is omitted during abstraction, how would it affect the performance of CAMAB algorithms? Can it be represented by IC error or reward discrepancy, and how?

**Q9 Complying With Reviewing Instructions:**

Yes

---

> ### Author Rebuttal · Authors · 2024-04-05
>
> We thank the reviewer for their time and insightful comments. We offer some further discussion in the following.
>
> > Most of their theoretical results are corollaries of similar results in CMAB. The conditions of these theoretical results should be illustrated in example or experiments for justification.
>
> Regarding the technical novelty of theoretical results, we refer the reviewer to our reply to reviewer YPfN. We agree that our theoretical results could be better demonstrated by experiments and examples. In Appendix E, we provide further scenarios that highlight the failure modes of each algorithm, but do not discuss how these examples relate to our theoretical regret bounds. We will provide further discussion in Appendix E elaborating on the connection between these examples and our theoretical results.
>
> > Some of definitions are a bit unclear: the procedure and target of CAMAB "problem" is not clearly given. Do the three types of CAMAB settings in section 5 aim for the same problem?
>
> We are grateful to the reviewer for highlighting this point. We will offer a better explanation and intuition of the procedure and the target of the CAMAB problem in Section 3, especially through the illustrative example 3.2. In the introduction of Section 5 where we delineate a taxonomy of CAMAB settings, we will also better clarify how our solutions fit in this taxonomy.
>
> > If there is a common cause of treatment and outcome before abstraction, and it is omitted during abstraction, how would it affect the performance of CAMAB algorithms? Can it be represented by IC error or reward discrepancy, and how?
>
> Changes in the causal structure during abstraction are captured by both the IC error \emph{and} the reward discrepancy. This is best illustrated by an example. Consider the scenario suggested by the reviewer in which a confounding $X$ variable, with direct descendants $Y$ and $Z$, is simply removed in the abstract model. In particular, consider the following base model:
>     \begin{align*}
>         X &= \text{Bern}(0.5) \\\\
>         Y &= 0.5X + \text{Bern}(0.1) \\\\
>         Z &= 0.5X + \text{Bern}(0.8),
>     \end{align*}
>     where each Bernoulli variable is independent. We may construct the following abstraction in which $X$, $Y$ and $Z$ are respectively mapped to $A$, $B$ and $C$:
>     \begin{align*}
>         \alpha_{A}(X) &= X\\\\
>         \alpha_{B}(Y)& =
>         \begin{cases}
>             2 & \text{if } Y > 0.5 \\\\
>             0 & \text{otherwise}
>         \end{cases}, \\\\
>         \alpha_{C}(Z) &=
>         \begin{cases}
>             2 & \text{if } Z > 0.5 \\\\
>             0 & \text{otherwise}
>         \end{cases}
>     \end{align*}
>
> Now consider the following trivial abstract model:
>     \begin{align*}
>         A &= \text{Bern}(0.5) \\\\
>         B &= 2 \cdot\text{Bern}(0.1)\\\\
>         C &= 2 \cdot \text{Bern}(0.8).
>     \end{align*}
>
> It is straightforward to check that the IC error between the base and abstract model is zero. This is surprising as $X$ has causal influence on $Y$ (and $Z$) but $A$ has no causal influence on $B$ (and $C$). However, this is possible as the abstraction $\alpha$ groups together values for $Y$ (and $Z$) so that the causal influence of $X$ becomes irrelevant. Now, assume that $Y$ represents a reward variable in a CMAB. Since the abstraction $\alpha$ maps different reward values of $Y$ to the same value of $B$ in the abstract model, the reward discrepancy will be non-zero. That is, the reward discrepancy captures the causal influence of $X$ on $Y$ that is lost due to the grouping together of reward values.
>
> Next, consider the following trivial abstraction:
>     \begin{align*}
>         \alpha_{A}(X) &= 0\\\\
>         \alpha_{B}(Y)& = Y  \\\\
>         \alpha_{C}(Z) &= Z,
>     \end{align*}
>     and the following abstract model:
>     \begin{align*}
>         A &= 0 \\\\
>         B &= 0.5\cdot \text{Bern}(0.5) + \text{Bern}(0.1) \\\\
>         C &= 0.5 \cdot \text{Bern}(0.5) + \text{Bern}(0.8)
>     \end{align*}
> In this case, the reward discrepancy is zero. However, note that $B$ and $C$ are always independent, whilst $Y$ and $Z$ are always dependent. As a result, the IC error is non-zero and captures the causal influence of $X$ on $Y$ and $Z$.

---

### Official Review · Reviewer_YPfN · 2024-03-22

**Q2-1 Originality-Novelty:** 2
**Q2-2 Correctness-Technical Quality:** 3
**Q2-5 Clarity Of Writing:** 3

**Q1 Summary And Contributions:**

This paper introduces a framework for transferring information between causal multi-armed bandits based on causal abstraction. It explores three scenarios, proposes algorithmic solutions for each, and evaluates their behavior regarding regret.

**Q2-3 Extent To Which Claims Are Supported By Evidence:**

3: Good: the main claims are supported by convincing evidence (in the form of adequate experimental evaluation, proofs, (pseudo-)code, references, assumptions).

**Q2-4 Reproducibility:**

3: Good: key resources (e.g. proofs, code, data) are available and key details (e.g. proofs, experimental setup) are sufficiently well-described for competent researchers to confidently reproduce the main results.

**Q3 Main Strengths:**

The paper explores an interesting problem in decision-making with MABs by proposing a framework for causally abstracted MABs, focusing on transfer learning across related problems with different variables and abstraction maps.

**Q4 Main Weakness:**

The paper's theoretical contributions are somewhat incremental, with results appearing as straightforward extensions or corollaries of existing work in the causal multi-armed bandit literature.

**Q5 Detailed Comments To The Authors:**

It would be beneficial if the authors could clarify how the inclusion of transfer learning into their framework enhances regret bounds compared to traditional approaches.

**Q9 Complying With Reviewing Instructions:**

Yes

---

> ### Author Rebuttal · Authors · 2024-04-05
>
> We are grateful to the reviewer for their comments and suggestions. In the following we address some of the raised points.
>
> > The paper's theoretical contributions are somewhat incremental, with results appearing as straightforward extensions or corollaries of existing work in the causal multi-armed bandit literature.
>
> Whilst we agree that our work leverages existing techniques and results in the bandit literature, we argue that this is aligned with the contributions of our work. More specifically, the goal and contribution of our paper are to highlight the subtle nuances and pitfalls that arise when combining abstraction with bandit algorithms. In this sense, our work is similar in spirit to [Barenboim et al. 2015], who investigate how the presence of unobserved confounders effects the performance of classical bandit algorithms. Our theoretical results aim to characterise the regret of abstraction-based extensions to classical bandit algorithms -- in terms of IC error and reward alignment, and provide a clear picture of how such extensions may fail (or succeed) to provide sublinear regret guarantees. As such it is only natural that our analysis relies heavily on existing techniques, and roughly corresponds to ``pushing'' existing regret guarantees through the abstraction map $\alpha$.
>
> > It would be beneficial if the authors could clarify how the inclusion of transfer learning into their framework enhances regret bounds compared to traditional approaches.
>
> As the reviewer keenly observes, CAMABs possess many similarities to transfer learning problems. In short, the presence of an abstraction implies that the interventional distributions of the base CMAB and abstracted CMAB are closely related. Exploiting this relationship is similar to exploiting task similarity in transfer learning. For example, the \texttt{IMIT} algorithm we propose is similar to the use of offline reinforcement learning policies in transfer learning. We stress that our work fundamentally differs from previous lines of work connecting transfer learning and causal inference that assume a fixed causal graph between tasks [Rojas-Carulla et al. 2018, Pearl et al. 2011]. The formalism of causal abstractions allows us to transfer information between learning problems even when the underlying variables and causal graphs of two tasks differ. We will make this point more salient in the paper.

---

### Official Review · Reviewer_GQ5q · 2024-03-23

**Q2-1 Originality-Novelty:** 3
**Q2-2 Correctness-Technical Quality:** 3
**Q2-5 Clarity Of Writing:** 3

**Q1 Summary And Contributions:**

This paper studies transfer learning with causal multi-armed bandits (CMAB) by introducing the causally abstracted MAB. The authors proposed an algorithm to learn the abstraction and analysis of the regrets.

**Q2-3 Extent To Which Claims Are Supported By Evidence:**

2: Fair: the main claims are somewhat supported by evidence (but the experimental evaluation may be weak, or does not match entirely with the claims, important baselines may be missing, proofs contain important ideas but lack rigor, algorithmic details are only discussed superficially, references are imprecise, assumptions are not sufficiently motivated or explicated, etc.).

**Q2-4 Reproducibility:**

3: Good: key resources (e.g. proofs, code, data) are available and key details (e.g. proofs, experimental setup) are sufficiently well-described for competent researchers to confidently reproduce the main results.

**Q3 Main Strengths:**

This paper provides an interesting and novel perspective on transfer learning in the MAB problem.

The problem is well-motivated and the paper structure is clear and easy to follow.

The results are derived rigorously in the paper.

**Q4 Main Weakness:**

The proposed method only evaluated with simulated data.

**Q5 Detailed Comments To The Authors:**

Does the proposed method's performance relate to the type of data shifting between different environments?

Does the proposed method assume causal sufficiency on the observed variable?

**Q9 Complying With Reviewing Instructions:**

Yes

---

> ### Author Rebuttal · Authors · 2024-04-05
>
> We thank the reviewer for the their time and their feedback. We address here some of the comments offered.
>
> > The proposed method only evaluated with simulated data.
>
> We acknowledge that our methods are evaluated on simulated data; this follows the standard of the bandit literature. Evaluation on real-world data presents a non-trivial challenge: our methods require fully specified models of CMABs at different levels of abstraction in order to run simulations, and such models are rarely publicly available. Still, we provided an evaluation relying on the more complex and realistic model proposed in the literature by [Lu et al. 2020], based on Adobe advertisement data. We recognize that the definition and evaluation of more realistic models, as well as the development of algorithms relying only on data, are relevant directions for future work.
>
> > Does the proposed method's performance relate to the type of data shifting between different environments?
>
> The performance of our methods is related to how well the abstraction map relates the distributions (and therefore the shift) of a base model and an abstracted model. We quantified this difference in the measures of interventional consistency error and reward discrepancy. Notice that, by defining a new abstracted model, an abstraction map can induce significant data shifts, differently from an intervention that can induce data shift only by changing a mechanism.
>
> > Does the proposed method assume causal sufficiency on the observed variable?
>
> In our setup, we do not need to assume causal sufficiency. Following the abstraction framework of [Rischel 2020], our results hold for semi-Markovian models which allow for unobserved confounders. Our simulations dealt with Markovian models, but the same results would hold for semi-Markovian models; in such a case, standard UCB would be better substituted by a bandit algorithm aware of confounding (for instance, [Bareinboim et al., 2015]).

---

### Meta-Review · Area_Chair_25hD · 2024-04-21

The paper discusses how to transfer information and policies across different Structural Causal Models (SCMs), defined on possibly different variables, in a Multi-Armed Bandit (MAB) decision making problem. The authors formalize the problem by assuming that one of the two models is an inexact Causal Abstraction of the other. This allows them to study the error and the regret of computing a solution for the abstract MAB on the lower-level SCM. Overall, the paper highlights the required properties of the abstraction map and that transferring information improves the regret performance compared to directly solving the problem on the abstract model. There has been a robust discussion.